# Generating Hypotheses of Dynamic Causal Graphs in Neuroscience: Leveraging Generative Factor Models of Observed Time Series

**Zachary C. Brown** [1]  **David Carlson** [1 2]

## Abstract

The field of hypothesis generation promises to reduce costs in neuroscience by narrowing the range of interventional studies needed to study various phenomena. Existing machine learning methods can generate scientific hypotheses from complex datasets, but many approaches assume causal relationships are static over time, limiting their applicability to systems with dynamic, state-dependent behavior, such as the brain. While some techniques attempt dynamic causal discovery through factor models, they often restrict relationships to linear patterns or impose other simplifying assumptions. We propose a novel method that models dynamic graphs as a conditionally weighted superposition of static graphs, where each static graph can capture nonlinear relationships. This approach enables the detection of complex, time-varying interactions between variables beyond linear limitations. Our method improves f1-scores of predicted dynamic causal patterns by roughly 22-28% on average over baselines in some of our experiments, with some improvements reaching well over 60%. A case study on real brain data demonstrates our method's ability to uncover relationships linked to specific behavioral states, offering valuable insights into neural dynamics.

## 1. Introduction

Causal discovery methods applied to brain dynamics hold great promise for designing targeted interventions in various neurological diseases. A typical approach is to generate hypotheses about causal relationships between brain

---

[1]Department of Electrical and Computer Engineering, Duke University, Durham, NC 15213, USA [2]Departments of Civil and Environmental Engineering; Biostatistics and Bioinformatics; and Computer Science, Duke University, Durham, NC 15213, USA. Correspondence to: Zachary C. Brown <zac.brown@duke.edu>, David Carlson <david.carlson@duke.edu>.

*Proceedings of the $42^{nd}$ International Conference on Machine Learning*, Vancouver, Canada. PMLR 267, 2025. Copyright 2025 by the author(s).

regions by using Granger causality on time series recorded from each region (Seth, 2007), which can be extended to many brain regions to capture more complete network relationships. Granger causality describes whether the history of one variable can help predict future information about another variable (Löwe et al., 2022; Assaad et al., 2022). Granger causal models are more statistical in nature than causal (Pearl, 2009); however, we seek to generate *hypotheses* about causal relationships to *then* inform causal inference / assays, hence the use of Granger causal models.

Current machine learning (ML) methods do not capture the brain's full complexity, which has well-documented nonlinear interactions and dynamic causal graphs that vary by state and task. Existing discovery methods tend to assume that either *a)* (potentially dynamic) linear models are sufficient for modeling causal behavior (Huang et al., 2019; Gallagher et al., 2021) or that *b)* causal graphs between variables are static and not state-dependent (Zhang et al., 2017b; Sadeghi et al., 2024). These limitations reflect a gap in existing ML methodology, with implications outside neuroscience for any field seeking to automate hypothesis generation for potentially nonlinear, state-dependent dynamical systems.

Thus, we focus on generating hypotheses about mechanisms governing nonlinear *and* state-dependent (or "dynamic") behavior in time series systems (hereafter referred to as "temporal systems" or simply "systems" to conserve space). While understandable concerns may be raised about the detectability of such mechanisms in even simple systems (see our discussion on identifiability in Appendix A.1), prior literature has demonstrated that using automated hypothesis generation for these mechanisms can still yield useful insights which aid scientific discovery (Mague et al., 2022). Indeed, we accurately estimate causal graphs in observed systems without making restrictive assumptions as to the state-dependence or linearity/nonlinearity of causal relationships. We do this by combining key concepts from both factor-based and nonlinear models of causal time series.

In summary, we seek to reduce the space of hypotheses that must be tested before a causal relationship is successfully identified. As such, we present 1) a novel approach to hypothesis generation for dynamic causal graphs using a deep generative factor model, 2) methods for including auxiliary

variables - such as behavior - as supervised labels to improve hypotheses and utility, 3) neuroscience-inspired relational discovery challenges and datasets for improving ML methods, and 4) empirical results comparing leading methods on under-explored hypothesis generation paradigms. We only assume our data consists of regularly sampled time series of scalar-valued nodes (see Section 3.1). These time series may be noisy, and we make no assumptions regarding the underlying generative processes. In Section 3.5, we assume the availability of "global state" labels for each time step that relate to the dominant generative process.

## 2. Related Work

Causal discovery may be considered to be a sub-field of hypothesis generation, and is the sub-field most closely related to our aim of forming hypotheses about causal relationships. Specifically, our proposed method should be viewed as a preparatory analysis technique preceding more complex methods such as Dynamic Causal Models (DCMs) (Friston et al., 2003). This is because we assume we have no access to any interventional inputs required to construct a DCM model (Stephan et al., 2010). In such a setting, it is recommended that Granger causal models (such as ours) be used to inform DCM models (Friston et al., 2013).

There are numerous causal discovery methods which avoid Granger causality (Runge, 2020; Gerhardus & Runge, 2020; Zou & Feng, 2009; Yu et al., 2019). These methods do not assume causality from temporal correlation, and thus are better positioned to handle causal relations such as contemporaneous effects (Runge, 2020) and latent confounders (Gerhardus & Runge, 2020). However, these methods tend to rely on assumptions that rule out hypotheses we would like to explore in neuroscience; for example, many methods assume causal graphs are acyclical (Runge, 2020; Gerhardus & Runge, 2020; Zou & Feng, 2009; Yu et al., 2019), which recurrent, multi-time scale systems like the brain may fundamentally violate - e.g., through cross-frequency coupling (Dzirasa et al., 2010; Allen et al., 2011). Thus, our approach differs from these methods and we instead report empirical comparisons between our method and Regime-PCMCI (Saggioro et al., 2020) in Section 4.2.

Vector Autoregressive Models (VARs) are often used to estimate Granger causality, but are limited to linear relationships (Shojaie & Fox, 2022). Recent research has used artificial neural networks (ANNs) to perform nonlinear causal discovery in temporal systems (Tank et al., 2021), including tasks from finding predictive graphical models (Löwe et al., 2022) to estimating a subject's emotional state via recorded electroencephalogram (EEG) signals (Song et al., 2018). These frameworks tend to focus on estimating static causal graphs (Tank et al., 2021; Song et al., 2018; Bussmann et al., 2021; Pamfil et al., 2020; D'Acunto et al., 2022) and we

use them as baselines in this paper. Our contributions are orthogonal to this line of work, as we focus on capturing dynamics of graphs over different points in time rather than estimating a static graph. In fact, our factor-based approach can use these structures as individual graphs whose strength modulates over time to make a composite graph.

There is also prior work on identifying dynamic graphs over time. For example, Fujiwara et al. (2023) developed an approach to estimate how the strength of causal relationships change over time, but did not capture dynamic graph structures. Other approaches - based on factor models or matrix factorizations - capture dynamic graph structures, but are restricted to discovering linear relationships between variables (Fox et al., 2011; Mague et al., 2022; Gallagher et al., 2021; Calhoun et al., 2014). Similarly, the very recent DyCAST method is tested solely on synthetic data with linear relationships (Cheng et al., 2025). Building on this work, we use factor model approaches on top of ANN base units to capture time-evolving graphs with nonlinear relationships. This allows us to more realistically capture key properties of many real-world temporal systems, including brain models. Additionally, our framework allows for the inclusion of auxiliary labels, similar to what has been done previously with systems limited to linear relationships (Talbot et al., 2023).

There is also work using dynamic nonlinear models through time, such as State-Dependent Causal Inference (SDCI) (Balsells-Rodas et al., 2022), Switching Neural Network Trackers (SNNTs) (Ghimire, 2018), and Constraint-based causal Discovery from Nonstationary/heterogeneous Data (CD-NOD) (Zhang et al., 2017a). While SDCI-related concepts inform our work, SDCI itself is incompatible with our use case because it models interactions between embedded latent variables (which are difficult to tie back to physical entities), each with their own state labels (a granularity which behavioral assays in neuroscience often lack). Meanwhile, SNNTs are probabilistic in nature instead of being designed for causal discovery and use discrete states instead of a superposition of factors used in our Section 4.3 experiments and the design of our proposed method. In contrast, the development of CD-NOD was largely motivated by the task of performing causal discovery on neurological (esp. fMRI) data, and it thus shares similarities with our work in much of the problem setup and certain assumptions. Still, there are enough differences between our approach and CD-NOD - such as our focus on autoregressive systems and unconstrained directionality of underlying causal relationships - that our direct comparisons with Regime-PCMCI (Saggioro et al., 2020) can be considered sufficient for our purposes.

Finally, prior work explored using low-dimensional latent spaces to model dynamics in neural activity (Linderman et al., 2017; Keshtkaran et al., 2022). In contrast, we largely ignore our models' latent features beyond applying con-

straints because our focus is on identifying candidates for causal relationships in recorded data.

## 3. Methods

### 3.1. Problem Statement

Suppose we must hypothesize a causal model for a dataset $\mathcal{D}$ of $N \in \mathbb{N}$ multi-dimensional (i.e., 'multi-channel') time series recordings $\mathbf{x}_n$. Let $\mathcal{D} = (\mathbf{X}) = (\mathbf{x}_n)_{n=1}^N$ denote the complete dataset of recordings. Each recording $\mathbf{x}_n \in \mathbf{X}$ contains the observed history of $n_c$ system variables at regular intervals with $T \in \mathbb{N}$ temporal measurements. Let $\tau_{\text{in}} \in \mathbb{N}_{\leq T}$ be the maximal number of time steps over which we estimate causal relationships. For simplicity, assume observations are of scalar features such that $\mathbf{x}_{n,t:t+\tau_{\text{in}}} \in \mathbb{R}^{n_c \times \tau_{\text{in}}} \ \forall \ t \leq T$ and $n \leq N$.

We assume our data is a noisy realization from a time-dependent system $\Phi(t)$. We must learn a model that *a)* reasonably approximates $\Phi$ at any point in time and *b)* estimates causal influences between system variables in $\Phi$. We approximate $\Phi$ by learning a factor model $f_\phi$ over the set of observed variables, where the expression of each factor is dynamic over time. Indeed, in Sections 4.1, 4.2, and 4.3 we implement $\Phi$ as a factor model itself (see Section 3.2 for formalization). The functions comprising each factor can take on many forms including hypothesis generation functions. Additionally, the linear combinations in factor models enable us to track statistics such as presence of a factor over time or in relationship to auxiliary variables (e.g., behavior), making these models useful for scientific research.

### 3.2. General Notation

Table 1 presents important symbols in this work. In particular, we use $\odot^\dagger$ to express 'broadcast multiplication' between vectors and 3-axis tensors, in which we take the Hadamard product (Kolda & Bader, 2009) between vector elements and corresponding matrix-shaped tensor slices before summing over all product results; see Appendix A.3 for details. Using this notation, our experiments treat $\Phi$ as

$$\Phi(t) = \mathcal{G}(t) \odot^\dagger \mathcal{F}(t) = \sum_{k=1}^{K^*} \mathcal{G}(t)_k \mathcal{F}_k(t)$$

where $K^*$ is the number of true factors $\mathcal{F}_k \in \mathcal{F}$ acting on all system variables $\mathcal{V}$ at each time step $t$ - i.e., $\mathcal{F}_k : \mathbb{R}^\mathcal{V} \to \mathbb{R}^\mathcal{V}$ - according to the system state $\mathcal{G}(t) \in \mathbb{R}^{K^*}$.

Our model consists of functions $f_{\phi_k}$ for each factor, each mapping a record of past variable states to present variable states. By predicting present variable states from time-lagged states of (other) variables, each $f_{\phi_k}$ learns Granger causal relationships between variables (Seth, 2007).

We now discuss how estimates of these relationships can be

*Table 1.* Common symbols in this paper.

| SYMBOL | DESCRIPTION |
|---|---|
| '^' | PREDICTION, FORECAST, AND/OR ESTIMATE |
| '$\odot^\dagger$' | 'BROADCAST MULTIPLICATION' (SEC. 3.2) |
| '$*$' | (EMPH.) SCALAR-VECTOR MULTIPLYING |
| '$\overline{(\ldots)}$' | AVERAGE VALUE OF $(\ldots)$ |
| 'MSE' | THE MEAN SQUARED ERROR PENALTY |
| 'COSSIM' | THE COSINE SIMILARITY MEASURE |

formally represented during evaluation. We turn to graph theory to express each factor's estimate of the strength of Granger causal relationships as a series of adjacency matrices mapping past variable (i.e., 'node') states/values to the present. The flexibility of graphs lets us also consider an abstract adjacency matrix summarizing the effect of $\Phi(t)$ on variables in the true system. Formally, we summarize our models' estimated Granger causal relationships in a series of adjacency matrices $\hat{\mathbf{A}} \in \mathbb{R}^{n_c \times n_c \times \tau_{\text{in}}}$, where each $a_{i,j,t} \in \hat{\mathbf{A}}$ is the estimated weight of the Granger causal relationship where the state of node $\nu_j \in \mathcal{V}$ from $t$ time steps in the past 'causally' effects current node $\nu_i \in \mathcal{V}$. To make quantitative comparisons, we standardized the representation(s) of baselines' and our models' estimated causal graph(s) by summing over time-lagged features (see Appendix B.2); hence, we define the lag-summed view of $\hat{\mathbf{A}}$ as

$$\tilde{\mathbf{A}} = \sum_t \hat{\mathbf{A}}_{[:,:,t]}. \tag{1}$$

Importantly, $\hat{\mathbf{A}}$ and $\tilde{\mathbf{A}}$ are summary statistics computed separately from the forecasting operation, and do not imply any linearization assumption in the forecasts themselves.

### 3.3. Base (Single-Factor) Model

Consider the case where our factor model is a single factor function $f_{\phi_1}$. We can use any algorithm for $f_{\phi_1}$ so long as it learns relevant Granger causal information by forecasting time series. In this paper we implement $f_{\phi_1}$ as a cMLP (Tank et al., 2021) due to its versatile design. Formally, the forward computation is given as

$$f_{\phi_1}(\mathbf{x}_{n,t:t+\tau_{\text{in}}}) = \hat{\mathbf{x}}_{n,t+(\tau_{\text{in}}+1)}. \tag{2}$$

Regarding the loss function of this single-factor model, we assume no direct knowledge of the target system's underlying causal graph, since we are performing hypothesis generation. Instead, we estimate causal connections in an *unsupervised* fashion. Hence, our objective function is essentially a sum over regularization terms guiding the model towards solutions with desirable qualities, like sparsity.

Formally, suppose we are given a recorded signal $\mathbf{x}_n \in \mathbf{X}$, the model's corresponding forecast $\hat{\mathbf{x}}_n \in \hat{\mathbf{X}}$, and an

estimated adjacency matrix $\hat{\mathbf{A}} \in \phi$ formed via a subset of the model's parameters $\phi$. Our implementation of $f_\phi$ learns $\hat{\mathbf{A}}$ in the first layer of weights, which together take on the shape $(n_c \times n_c \times \tau_{in})$ and can be interpreted as a series of time-lagged adjacency matrices (Tank et al., 2021); details in Appendix B.1. Hence, we include a lag-dependent $L1$-norm sparsity penalty on the time-lagged slices of $\hat{\mathbf{A}}$ (favoring the discovery of temporally-local interactions in cases of cyclical or chained causal relationships in the factor) along with a MSE loss over forecasts in our base objective

$$\mathcal{L}_\beta(\phi, \mathcal{D}) = \eta \sum_{t=1}^{\tau_{in}} \log(t+1)||\hat{\mathbf{A}}_{:,:,t}||_1 + \omega \mathrm{MSE}(\mathbf{X}, \hat{\mathbf{X}})$$
(3)

where $\eta, \omega \in \mathbb{R}$ are chosen hyperparameters. Again, the $L1$ regularization term here serves two purposes for the factor's causal model: 1) to encourage sparsity/parsimony, and 2) to emphasize reliance on more recent time lags via the log term. We use the Adam optimization algorithm to train our model to minimize the above loss function. We also include weight decay (i.e., $L2$-norm regularization) on all parameters as done in prior work (Tank et al., 2021).

### 3.4. RElational Discovery via ConditionaLly Interacting Forecasting Factors (REDCLIFF)

Now consider general cases where our model has $n_k \geq 1$ factors. We seek to learn a Granger causal model $f_\phi$ satisfying $f_\phi(\mathbf{x}_{n,t:t+\tau_{in}}) = \sum_{k=1}^{n_k} \alpha_{n,k,t+(\tau_{in}+1)} f_{\phi_k}(\mathbf{x}_{n,t:t+\tau_{in}})$ where $n_k \in \mathbb{N}$ is the total number of model factors, $f_{\phi_k}$ is the $k$-th factor generator, and $\alpha_{n,k,t+(\tau_{in}+1)} \in \mathbb{R}$ is the coefficient for $f_{\phi_k}$ at time $t + (\tau_{in} + 1)$.

To learn each factor coefficient, we introduce a parameterized function $g_\theta$ (the 'state model') which generates factor weights (or 'scores') conditioned on the system's history (and nothing else; see Balsells-Rodas et al. (2022)'s "state-determined" setting). Note $g_\theta$ can be seen as classifying each factor's importance in forecasts. Thus, $g_\theta$ is fundamentally different from the generative factors of $f_\phi$ (which perform regression on the time series itself); indeed, we found that $g_\theta$ typically required more historical information than the generative factors to effectively learn their tasks (see Section 3.6 and Appendix A.2). Thus, we provide additional historical information of length $\tau_{cl} \in \mathbb{N}_{\leq T}$ to $g_\theta$ so $g_\theta$ sees a combined context window of $\tau_{in} + \tau_{cl}$ time steps.

We define the forward computation of $g_\theta$ according to $g_\theta(\mathbf{x}_{n,t-\tau_{cl}:t+\tau_{in}}) = \boldsymbol{\alpha}_{n,:,t+(\tau_{in}+1)}$ where $\boldsymbol{\alpha}_{n,:,t+(\tau_{in}+1)} \in \mathbb{R}^{n_k}$ are conditional factor scores. Thus, our model forecasts $\Phi$'s evolution as:

$$\hat{\mathbf{x}}_{n,t+\tau_{in}+1} = g_\theta(\mathbf{x}_{n,t-\tau_{cl}:t+\tau_{in}}) \odot^\dagger f_\phi(\mathbf{x}_{n,t:t+\tau_{in}}) \quad (4)$$

$$= \sum_{k=1}^{n_k} \alpha_{n,k,t+(\tau_{in}+1)} f_{\phi_k}(\mathbf{x}_{n,t:t+\tau_{in}})$$

where we have the $n_k$ factor scores $\alpha_{n,k,t+(\tau_{in}+1)} \in \mathbb{R} \; \forall \; k$ being broadcast-multiplied against the $n_k$ factor outputs $f_{\phi_k}(\mathbf{x}_{n,t:t+\tau_{in}}) \in \mathbb{R}^{n_c \times 1}$. We refer to the resulting model as a RElational Discovery via ConditionaLly Interacting Forecasting Factors (REDCLIFF) model (see Figure 1).

The objective function for training REDCLIFF models has two main components. Modifying Equation 3, the first component $\mathcal{L}_f$ pertains to the generative factors $f_{\phi_k}$ and retains the MSE forecasting penalty, but now includes all $n_k \geq 1$ factors in the $L1$-norm penalty. We also include a cosine similarity penalty to induce our prior that each factor should serve different purposes by encouraging factors to dissociate (details in Appendices B.3 and C.6). Specifically,

$$\mathcal{L}_f(\phi, \mathcal{D}) = \eta \sum_{k=1}^{n_k} \sum_{t=1}^{\tau_{in}} \log(t+1)||^k\hat{\mathbf{A}}_{:,:,t}||_1 \quad (5)$$

$$+\omega \mathrm{MSE}(\mathbf{X}, \hat{\mathbf{X}})$$

$$+\rho \sum_{p=1}^{n_k} \sum_{q=(p+1)}^{n_k} \mathrm{CosSim}(^p\tilde{\mathbf{A}} - \mathbf{I}, ^q\tilde{\mathbf{A}} - \mathbf{I})$$

where $\mathbf{I} \in \mathbb{R}^{n_c \times n_c}$ is the identity matrix and $^p\tilde{\mathbf{A}}$ and $^q\tilde{\mathbf{A}}$ are the lag-summed views of the adjacency matrices derived from factors $p$ and $q$ (Eq. 1).[1] We set $\eta$, $\omega$, and $\rho$ as real-valued hyperparameters. The second loss component $\mathcal{L}_g$ enforces sparsity (offset by 1) in the factor scores of $g_\theta$,

$$\mathcal{L}_g(\theta, \mathcal{D}) = \gamma\Big(-1 + \sum_{n=1}^{N} ||\boldsymbol{\alpha}_{n,:}||_1\Big), \quad (6)$$

where $\gamma \in \mathbb{R}$ is another constant hyperparameter. Putting it together gives the full objective function:

$$\mathcal{L}(\phi, \theta, \mathcal{D}) = \mathcal{L}_f(\phi, \mathbf{X}, \hat{\mathbf{X}}) + \mathcal{L}_g(\theta, \mathcal{D}). \quad (7)$$

In review, the state model $g_\theta$ is how a REDCLIFF model adjusts weights of relationships between variables. By emphasizing factors with different relationships, $g_\theta$ can model changes in the causal direction of relationships, making it the main function for estimating dynamic causal behavior.

### 3.5. REDCLIFF-Supervised (REDCLIFF-S)

Now suppose $\mathcal{D}$ includes a set $\mathbf{Y}$ of "global state" (i.e., "behavioral state" in our motivating neurobehavioral cases) labels for the span of each recording in $\mathbf{X}$; that is, $\mathcal{D} = (\mathbf{X}, \mathbf{Y})$. We assume the set $\mathbf{Y}$ takes the form $\mathbf{Y} = (\mathbf{y}_n \in \mathbb{R}^{B \times T})_{n=1}^{N}$ where $B \in \mathbb{N}$ is the number of global state variables and both $N$ and $T$ are as before.

---

[1]Subtracting the identity induces an implicit assumption that each variable is involved in each factor (via self-connection); this is reasonable for recurrent systems like the brain and in our Synthetic Systems experiments. Future work should explore alternatives.

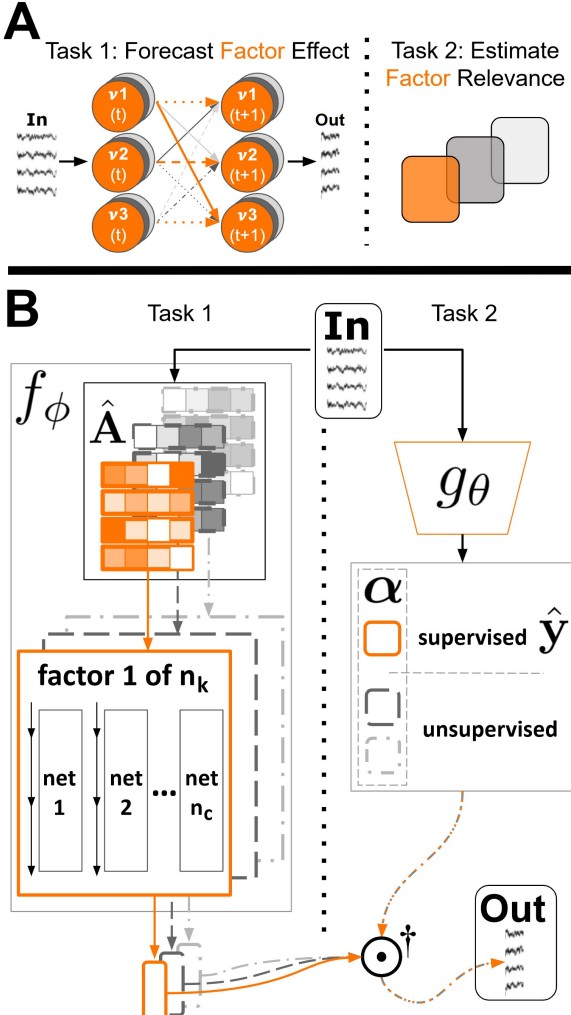

*Figure 1.* Illustration of the REDCLIFF(-S) algorithm. A) The two primary subtasks performed by a REDCLIFF(-S) model. B) An illustration of a $n_k = 3$ factor REDCLIFF-S model's forward pass.

We seek to estimate causal patterns linked to specific behaviors in signals. Thus, we follow prior literature and assign behaviors to the first $B$ elements of $\boldsymbol{\alpha}$ (Mague et al., 2022). To enforce better separation between the factors $f_{\phi_k}$ and state model $g_\theta$ and because our behavioral labels may be noisy, we define invertible functions $g_\alpha$ and $g_y$ such that $\hat{\mathbf{y}} = g_y(g_\alpha^{-1}(\boldsymbol{\alpha}_{:B}))$. Invertibility is necessary for fully preserving information between these operations, allowing us to deduce the relationship between label predictions and factor weights; see discussion in Appendix A.2 for further details. When the behavioral label $\mathbf{y}_n$ is a known quantitative variable, we can generate $\hat{\mathbf{y}}_n$ via supervised regression. Alternatively, if $\mathbf{y}_n$ is categorical, we can predict a behavior as 'present' in the signal when the value of one of these behavioral elements crosses a threshold $c_b \in \mathbb{R}$.

We put this all together by re-defining $g_\theta$ such that

$$g_\theta(\mathbf{x}) = \left[ \begin{array}{c} g_\alpha(g_h'(\mathbf{x})) \\ g_y(g_h'(\mathbf{x})_{:B}) \end{array} \right] = \left[ \begin{array}{c} g_\alpha(\boldsymbol{\alpha}') \\ g_y(\boldsymbol{\alpha}'_{:B}) \end{array} \right] = \left[ \begin{array}{c} \boldsymbol{\alpha} \\ \hat{\mathbf{y}} \end{array} \right] \tag{8}$$

where $g_\alpha : \mathbb{R}^{n_k} \to \mathbb{R}^{n_k}$ and $g_y : \mathbb{R}^B \to \mathbb{R}^B$ are invertible and now $g_h'$ represents the original function $g_\theta$ prior to our inclusion of $g_\alpha$ and $g_y$. The predicted behavioral label $\hat{y}_b \in \hat{\mathbf{y}}$ indicates the presence of behavior $y_{n,b}$ if $\hat{y}_{n,b} > c_b$ where $c_b$ is some threshold. We make no change to the REDCLIFF forward computation (Equation 4) other than to note the output of $g_\theta$ now includes $\hat{\mathbf{y}}$, which must be ignored in the broadcast multiplication step so just $\boldsymbol{\alpha}$ is used.

To complete the definition of a REDCLIFF-S model, we add a supervised component to $\mathcal{L}$ in Eq. 7 to obtain

$$\mathcal{L}(\phi, \theta, \mathcal{D}) = \mathcal{L}_f(\phi, \mathcal{D}) + \mathcal{L}_g(\theta, \mathcal{D}) + \lambda \text{MSE}(\mathbf{Y}, \hat{\mathbf{Y}}) \tag{9}$$

where $\lambda$ is a real-valued hyperparameter. We hypothesize this new MSE term constrains the Granger causal graph(s) of the relevant factor(s) to correspond to $\hat{\mathbf{Y}}$ (as would any other supervised loss term).

### 3.6. Training Strategies and Details

We train REDCLIFF(-S) using the Adam optimizer with weight decay. To effectively learn each model, we first pre-train the state model $g_\theta$ using the terms $\mathcal{L}_g$ and $\lambda \text{MSE}(\mathbf{Y}, \hat{\mathbf{Y}})$ from Equation 9 (setting $\lambda = 0$ in the unsupervised case). Then, we freeze the state model and train the generative factors in $f_\phi$ using Equation 5 (we call this "acclimation"). Following these steps, we employ the full training objective (Equation 7 or 9) to jointly update all parameters.

One challenge for hypothesis generation is that you cannot perform model selection explicitly using holdout graph discovery performance on real data as the true graph is unknown. Instead, we stop training once the number of maximal iterations is reached or when the following criterion is minimized on the validation set $\mathcal{D}_v = (\mathbf{X}_v, \mathbf{Y}_v)$:

$$\mathcal{L}_s(\phi, \theta, \mathcal{D}_v) = \overline{\omega \text{MSE}(\mathbf{X}_v, \hat{\mathbf{X}}_v)} + \overline{\lambda \text{MSE}(\mathbf{Y}_v, \hat{\mathbf{Y}}_v)} \tag{10}$$

$$+ \rho \overline{\sum_{p=1}^{n_k} \sum_{q=(p+1)}^{n_k} \text{CosSim}\left( \frac{{}^p\tilde{\mathbf{A}}}{\max({}^p\tilde{\mathbf{A}})}, \frac{{}^q\tilde{\mathbf{A}}}{\max({}^q\tilde{\mathbf{A}})} \right)}.$$

For more details on $\mathcal{L}_s$, see Appendix B.3.

In terms of computational complexity, the REDCLIFF(-S) implementations we tested in this work scale (in training and testing) as the product of the number of factors $n_k$ and the number of forward passes $n_{sim}$ made by each factor [2]

---

[2] We experimented with having $n_{sim} \in \mathbb{N}_{\geq 1}$ forward passes before updating REDCLIFF-S parameters in some grid searches. In practice, $n_{sim} = 1$ performed best, but we include the general form in our complexity analyses for rigor.

plus the computational complexity of the state model. More formally, let $C_{\phi_*}$ represent the *maximal* (temporal or spatial, depending on context) complexity of any factor in $\phi$ and $C_\theta$ be the complexity of the state model; then the temporal complexity of our implementation is $\mathcal{O}(n_k \cdot n_{sim} \cdot C_{\phi_*} + C_\theta)$ and the spatial complexity is the same. Learning can also be parallelized over factors for scalable implementations.

Additional theoretical analyses in Appendix A.2 support our findings that $g_\theta$ required more context than each $f_{\phi_k}$.

# 4. Experiments on Simulated Datasets

We now report empirical comparisons of REDCLIFF-S against other methods[3], with additional results given in Appendix C. All algorithms were trained using a local cluster that featured various GPU devices, including A6000 GPUs.

We focus much of our analysis on f1-score performance, partly because our neuroscientific application implicitly assumes all parts of the brain are causally related to some degree and the difficulty is determining which connections are the *most relevant* at key moments; the f1-score's emphasis on the presence of positive-label relationships is thus well suited to our setting. As is common practice, we report 'optimal' f1-scores in which we compute each evaluated algorithm's f1-score using a classification threshold (i.e., the value at which logits are labeled negative or positive) which yields the highest possible f1-score for that algorithm on the task. We do use the "Adjustment Identification Distance" (or AID) family of causal distance metrics (Henckel et al.) in our comparisons against several supervised baselines (Table 2 and Appendix C.3), since all algorithms in this comparison had access to 'global state' information related to causal edge relevancy, possibly making it harder for the f1-score to differentiate algorithms. We also report some ROC-AUC scores, as in Table 2 and the appendices.

## 4.1. Synthetic Data Generation and Analyses

We rely on synthetic data to test REDCLIFF-S' ability to estimate known causal relationships. To create these datasets, we defined each artificial temporal system as a set of Vector Auto-Regressive (VAR) models (Zivot & Wang, 2006), one for each factor. Our implementation allows for inter-variable edge activations between time-lags to be either linear or nonlinear; in this work, we included nonlinear activations between different time-lags. Specifically, we used the ReLU (that is $\text{ReLU}(x) = x\mathbf{1}_{[\max(x,0)]}(x)$) and negative ReLU (that is $\text{nReLU}(x) = x\mathbf{1}_{[\min(x,0)]}(x)$) nonlinear activations. For each experiment, we selected hyperparameters such as the number of VAR models used, how many time series were present, and how many lags were in each VAR (see

---

[3]Code has been made available at github.com/carlson-lab/redcliff-s-hypothesizing-dynamic-causal-graphs

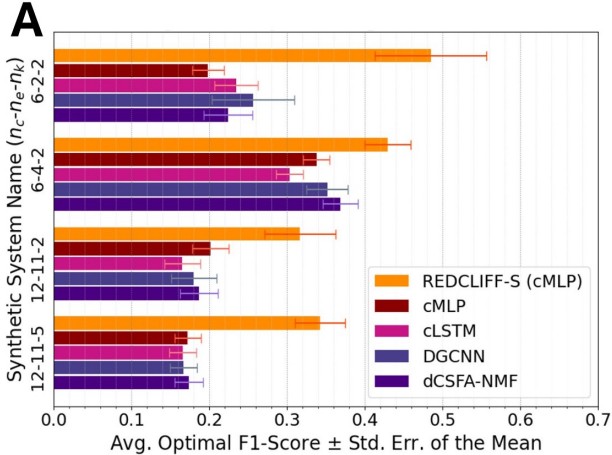

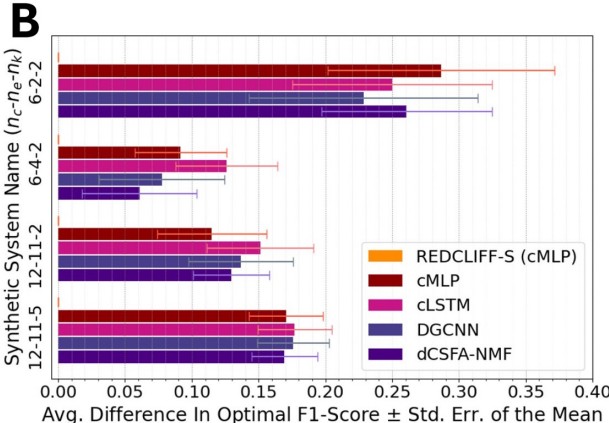

*Figure 2.* Synthetic Systems results from select systems. A) Average optimal f1-score and standard error of the mean (SEM) between true and estimated inter-variable causal relationships. B) Average pairwise improvement and SEM between optimal f1-scores obtained by REDCLIFF-S and baselines.

Appendix D). Here we set the number of lags for each VAR to $\tau = 2$ time steps for visualization purposes, but any integer greater than 0 would suffice. Defining these systems also meant selecting square adjacency matrices mapping values from past to present system states for each VAR model in the system. We included self-connections in each adjacency matrix to better visualize their generated time series. The number of inter-variable connections between time series was varied across systems, with connections chosen randomly. Our Synthetic Systems were designed to approximate the conditions of the TST dataset (Carlson et al., 2023) discussed in Section 5, with increasing fidelity / complexity (the largest systems had 12 nodes, which corresponded to 12 channels in our preprocessed TST dataset).

Once we defined a system, we sampled recordings and their

labels for the dataset. First, a random state vector was drawn from a uniform distribution for each VAR model in the system. This vector was recurrently fed through its corresponding VAR model for a number of 'burn-in' steps, after which we recorded the resulting multivariate time series for $T$ time steps; innovations were simulated throughout by multiplying draws from uniform and Gaussian distributions. Recordings obtained from each VAR model were then weighted over time with linearly interpolated weights (randomly selected between 0 and 1). These weighted recordings were added together along with a level of Gaussian noise. We augmented labels by marking with a one-hot vector which factor weight was largest at each time step, meaning *all of our Synthetic Systems experiments contain label noise*.

We stratified results by marking systems as low (L), moderate (M), or high (H) complexity $\mathfrak{C}$, according to

$$\mathfrak{C}(n_c, n_e) = r_{\text{graph}}^{-1} = \left( \frac{n_e}{(n_c{}^2 - n_c)} \right)^{-1} \quad (11)$$

where $r_{\text{graph}}$ is the ratio of true inter-variable interactions in each system state to the number of possible inter-variable interactions. In essence, $\mathfrak{C}$ is inversely related to the sparsity of relationships in the graph. We provide additional details regarding this complexity score in Appendices C and D.

## 4.2. Hypothesis Generation for Synthetic Systems

Figure 2 gives results from a high-complexity system (6-2-2) and three moderate system configurations for which we generated five random repeats (i.e., unique systems) to train each algorithm (more results in Appendix C). Each repeat was designed to have 1040 training samples and 240 validation samples for each factor/system state in the system, with each sample recording 100 sequential time steps. We used the cMLP and cLSTM (Tank et al., 2021); DGCNN (Song et al., 2018); and dCSFA-NMF (Talbot et al., 2023; Mague et al., 2022) methods as baselines. Hyperparameters for each algorithm were chosen via grid search on a single repeat in our (low-complexity) $n_c = 12$-node, $n_e = 33$-edge (inter-variable), $n_k = 3$-factor Synthetic System dataset. Our grid searches selected parameters which minimized stopping criteria (for DYNOTEARS and REDCLIFF-S) or objective functions (for all other baselines), with ties won by the most 'parsimonious' parameters (see Appendices D and E). The same hyperparameters were used across all the experiments reported in this section, with limited adjustments - e.g., to the number of REDCLIFF-S factors or the value of loss coefficients - made to account for certain characteristics in the target system (see Appendix E). Each algorithm was trained on a given system repeat dataset, after which we computed the average optimal f1-score and standard error of the mean (SEM) of each algorithm's inter-variable causal predictions against the true causal graphs across factors and repeats for each synthetic system.

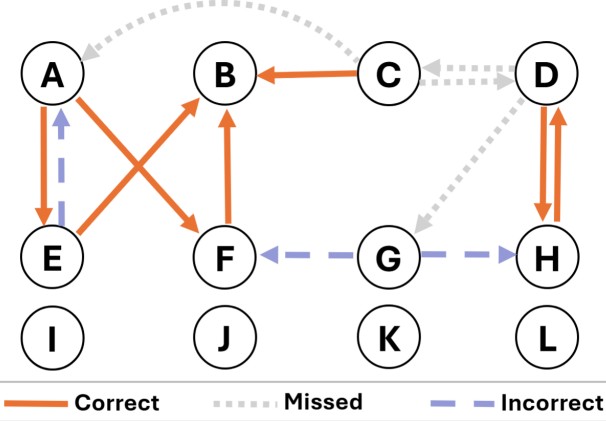

*Figure 3.* Visualizing true and REDCLIFF-S Top-10 estimated inter-variable causal relationships of a factor from Synthetic System "12-11-5". REDCLIFF-S captures 7 of 11 true relationships, while incorrect/missed pathways share functional similarities; the fact that REDCLIFF-S extracted these relationships with four other system states/factors adding noise in the dataset marks a sea change in modeling capability over baselines. While we could have highlighted examples with higher optimal f1-scores, we highlight this one due to its comparable complexity to our TST case study.

As another test, we re-trained REDCLIFF-S models along with Regime-PCMCI (Saggioro et al., 2020) and supervised versions of 'SLARAC', 'QRBS', and 'LASAR' from the tidybench repository (Weichwald et al., 2020) [4] on our '12-11-2' Synthetic Systems (Fig. 2A) using 10 different random seeds and compared their ability to detect unweighted causal edges. In Table 2, we report the average median performance of each method on several metrics, with additional details and results given in Appendix C.3.

## 4.3. D4IC Multi-State Extension of DREAM4

We also validated REDCLIFF-S on an adaptation of the public DREAM4 dataset (Marbach et al., 2009). We chose to adapt the DREAM4 dataset due to 1) it's use as a benchmark dataset in prior hypothesis generation and causal discovery research (Pamfil et al., 2020) and 2) its relatively small size since some baseline implementations required that data not be batched - namely, the DYNOTEARS algorithm (Pamfil et al., 2020) and the NAVAR algorithm (Bussmann et al., 2021) with recursive cLSTM-based (NAVAR-R) and cMLP-based (NAVAR-P) implementations. We measured the average optimal f1-score of each algorithm's causal estimates between variables, with causal relationships defined by factors taken from different folds of the DREAM4 dataset and presented to the algorithms in a single training session as

---

[4] We also tried training 'SELVAR' from tidybench, but could not compile it locally.

*Table 2.* Comparing unweighted inter-variable edge detection by supervised discovery methods on the 12-11-2 Synthetic Systems (Fig. 2) - note REDCLIFF-S' f1-scores differ from Fig. 2 which included edge weighting. Edge predictions were determined by optimal f1-score thresholds. Values report the mean (over repeats) of the median (over 10 random seeds and system factors) statistic $\pm$ SEM. 'Upper' and 'Lower' refer to upper- and lower-triangular portions of adjacency matrices, which we split to avoid cycles (see Sec. 4.1 & Appendix B.2). We also report the mean comparative placement (from 1 to 5) of each method's mean performance (details in Appendix C.3).

| | | (F1 Thresh.) ROC-AUC | Parent Aid Error | | Structural Hamming Dist. | | Mean |
|---|---|---|---|---|---|---|---|
| Algorithm | Optimal F1 | | Upper | Lower | Upper | Lower | Place. |
| LASAR (Sup.) | 0.344±0.034 | 0.579±0.018 | 7.0±1.995 | 4.3±0.460 | 21.2±5.410 | 20.9±5.905 | 3.5 |
| QRBS (Sup.) | 0.203±0.016 | 0.536±0.008 | 4.5±1.631 | 3.0±0.949 | 35.9±5.049 | 38.8±4.258 | 3.5 |
| SLARAC (Sup.) | 0.169±0.016 | 0.512±0.010 | 2.7±1.973 | 0.7±0.179 | 54.5±5.194 | 56.3±4.447 | 3.7 |
| Regime-PCMCI | 0.407±0.027 | 0.673±0.026 | 6.1±1.212 | 4.8±0.782 | 9.1±0.219 | 9.4±1.033 | 2.3 |
| **REDCLIFF-S** | 0.383±0.038 | 0.705±0.022 | 3.9±0.607 | 3.1±0.727 | 10.9±1.513 | 11.8±1.753 | **2.0** |

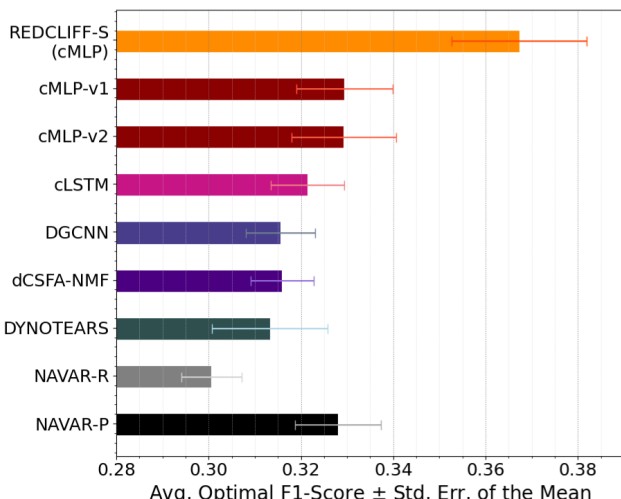

*Figure 4.* Average optimal f1-score between true and estimated inter-variable causal relationships across the D4IC HSNR dataset.

a single dynamic-causal system. Non-stochastic noise was simulated by adding down-weighted recordings from all but one DREAM4 folds to a recording from the dominant fold at three different signal-to-noise ratios: low (LSNR), moderate (MSNR), and high (HSNR). We refer to the resulting dataset as the "DREAM4 In Silico-Combined" (D4IC) dataset.

Hyperparameters for each algorithm were tuned to a single repeat of the D4IC MSNR dataset prior to training on all D4IC noise levels and repeats. Model selection proceeded as in Section 4.2, although we increased the weight on the cosine similarity penalty in selecting our final REDCLIFF-S model which balanced the scale of variation of the cosine similarity penalty with that of the other penalties (details in Appendix B.3). We include two cMLP baseline models (v1 and v2) with slightly different hyperparameters. Results are visualized in Figure 4, in which we see REDCLIFF-S improves over alternative approaches on this dataset.

### 4.4. Discussion

The main strength of REDCLIFF-S is that its hypotheses can be easily tested. Equations 4 and 8 are designed for this in several ways. As discussed, factor scores $\alpha_{n,k,t}$ can be given explicit behavioral designations. Factor scores can also be constrained (via Sigmoid or similar activation functions) to be nonnegative, and thereby be viewed as how (physically) present the corresponding factors are in predicted signals.[5] Thirdly, trained factors $f_{\phi_k}$ rely on fixed graphs explicitly showing relationships between variables. Finally, the sum over factors allows us to inspect individual contributions of each factor. *Overall, REDCLIFF-S shows performance improvements in many scenarios.*

First, Figure 2 and Table 2 together give the core results on our Synthetic Systems Dataset experiments. Figure 2A presents average optimal f1-scores (across all samples and repeats) of REDCLIFF-S and applicable baselines. Surprisingly, the single-factor baselines attain fairly competitive f1-scores in some cases, seemingly learning an 'average factor graph' that may not be particularly accurate for any single factor but which performs well 'on average'. Even so, in Figure 2B we see the average *pairwise* improvement of REDCLIFF-S' predictions is at least a standard error of the mean above zero for all baselines on all four systems, suggesting REDCLIFF-S yields improvements with statistical significance. These findings are further supported by the results shown in Table 2. Despite the fact that REDCLIFF-S was the only algorithm trained to predict global state information (all other baselines in Table 2 had direct access to the state label), REDCLIFF-S is uniquely able to achieve competitive performance across all metrics used. This can be seen in that when we rank the algorithms based on their mean performance for each metric, REDCLIFF-S' ranking (or 'comparative placement', to avoid confusion with 'matrix rank') tends to be lower than the other methods'. Taken together, our findings in Section 4.2 suggest REDCLIFF-S

---

[5]Our early experiments included these constraints, but we omit them here due to different models being selected for by our criteria.

*Table 3.* Summarizing effects of REDCLIFF-S ablations on mean optimal f1-score across various datasets (Appendix C.6).

| | ABLATION | | | |
| DATASET | $\rho = 0$ | $n_k = 1$ | $\alpha = 1$ | $\lambda = 0$ |
|---|---|---|---|---|
| SYN. SYS. 6-2-2 | ↓ | ↓ | ↓ | ↓ |
| SYN. SYS. 6-4-2 | ↑ | ↓ | ↓ | ↓ |
| SYN. SYS. 12-11-2 | ↓ | ↓ | ↓ | ↓ |
| SYN. SYS. 12-11-5 | ↑ | ↓ | ↓ | ↓ |
| D4IC HSNR | ↓ | ↓ | ↓ | ↓ |

generates hypotheses about the presence of dynamic causal relationships that prove accurate at state-of-the-art rates, and this pattern holds true for systems with differing numbers of variables, inter-variable relationships, and factors.

Results from our D4IC HSNR experiments are shown in Figure 4, where we see the REDCLIFF-S (cMLP) model achieves the highest average optimal f1-score with a statistically significant margin. While the improvements over competing methods are somewhat modest in this case, a brief investigation into the factor networks of D4IC HSNR suggests that four of the five factors in each repeat would have been rated as 'low complexity' according to Equation 11; importantly, REDCLIFF-S tends to make the least improvement over baselines in the 'low complexity' regime (see Appendix C.3). In all, our D4IC experimental results generally support those in Section 4.2, verifying that REDCLIFF-S is effective at hypothesizing dynamic causal relationships.

We also ran ablation tests to study the effect of REDCLIFF-S model parameters on performance, which we discuss in detail in Appendix C.6 and summarize in Table 3.

See further discussion in Appendix B.4.

## 5. Experiments on Real-World Datasets

We tested REDCLIFF-S on real-world data using the publicly available Tail Suspension Test (TST) dataset (Carlson et al., 2023) - which we discuss here - and the Social Preference (SP) dataset described in Mague et al. (2022) (see Appendix C.5). Where possible, we combined multiple local field potential (LFP) recordings within a single brain region by taking their average. The number of factors and other hyperparameters were chosen based on grid searches minimizing the stopping criteria for training. Appendix D covers TST preprocessing in detail, and Appendix C.4 elaborates on TST model selection and additional results.

In Figure 5 we show the difference between mean normalized causal estimates for the REDCLIFF-S factors assigned to the Open Field (OF) and Home Cage (HC) behavioral paradigms in this case study. The OF factor relies more heavily than the HC factor on information from the Nucleus

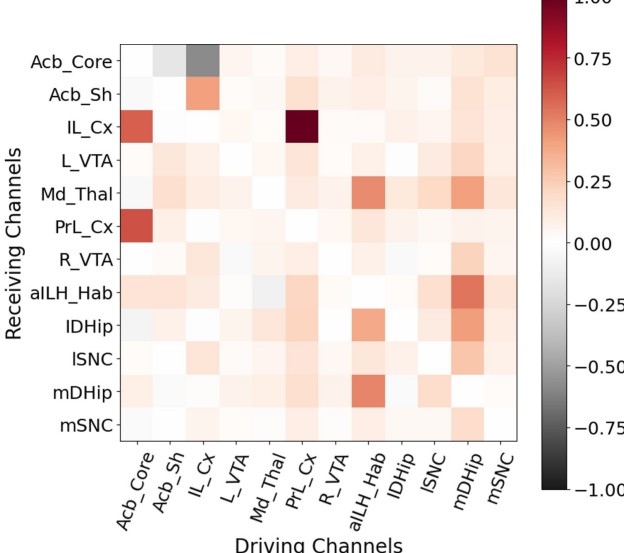

*Figure 5.* Difference in REDCLIFF-S' mean, normalized inter-variable causal estimates in the Open Field vs Home Cage paradigms of our TST Case Study.

Accumbens Core (Acb_Core) to predict behavior in the Prelimbic and Infralimbic cortices, and from the Anterior-Left Lateral Habenula ('alLH_HAB') and Medial Dorsal Hippocampal (mDHip) channels to predict activity in several regions, including the Thalamus (Md_Thal). Interestingly, our model's hypothesis that the Anterior-Left Lateral Habenula ('alLH_HAB') is important for predicting LFP activity across several brain regions in the more stressful OF paradigm appears to be validated by recent research implicating the Lateral Habenula in stress responses of mice (Tan et al., 2024). Previous research also identified a brain-wide 'anxiety network' involving the Nucleus Accumbens Shell and the Prelimbic and Infralimbic cortices (Talbot et al., 2023), which our REDCLIFF-S model appears to express, in part, as pathways from the Nucleus Accumbens Core to the Prefrontal Cortex and from there to the Infralimbic Cortex and then to the Nucleus Accumbens Shell.

## 6. Conclusion

We present the novel REDCLIFF-S hypothesis generation algorithm for temporal systems featuring dynamic causal interactions. By learning to combine nonlinear factors using weights conditioned on signal history, REDCLIFF-S attains state-of-the-art performance in estimating multiple causal graphs simultaneously in various settings and in a way conducive to follow-up scientific inquiry. Experiments on real data suggest that REDCLIFF-S can provide scientific value to computational neuroscience and other fields.

## Acknowledgements

We would like to thank our colleagues at the Collective for Psychiatric Neuro-Engineering (CPNE) for supporting this work. In particular, conversations with Noah Lanier, Aaron Fleming, Casey Hanks, Kathryn Walder, and Hannah Soliman proved especially helpful in clarifying and communicating our ideas. We also thank Carles Balsells-Rodas for his contributions during the early phases of this project, particularly his guidance on SDCI and related concepts which informed our work. Conversations with Natalie Brown were also useful for clarifying key ideas. Finally, we thank the reviewers for providing their insight and helpful feedback.

This material is based upon work supported by the National Science Foundation Graduate Research Fellowship under Grant No. DGE 2139754. Research reported in this publication was also supported by the National Institute Of Mental Health of the National Institutes of Health under Award Number R01MH125430. The content is solely the responsibility of the authors and does not necessarily represent the official views of the National Institutes of Health.

## Impact Statement

This paper presents work whose goal is to advance the field of Machine Learning. There are many potential societal consequences of our work, none which we feel must be specifically highlighted here.

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

# A. Supporting Theory and Proofs

## A.1. Identifiability Analysis of Systems Featuring Nonlinear, Dynamic Causal Graphs

In this section, we provide theoretical arguments as to why the systems at the center of our work - dynamical systems featuring nonlinear, dynamic causal graphs - are generally not identifiable. At the same time, we point out that prior work has demonstrated that using causal discovery methods can still benefit scientific research in real-world settings where nonlinear, dynamic causal graphs are the norm rather than the exception (Mague et al., 2022). Throughout our discussion, we follow the common definition of identifiability, in which if the likelihood of two models is the same, then their parameterizations must also be the same (Maclaren & Nicholson, 2019).

There are three cases related to causal identifiability in our setting: the case where auxiliary variables are not available (the unsupervised case), that where some - but not all - auxiliary variables are available (the partially supervised case), and the case where an auxiliary variable can be assigned to every system state (the fully supervised case). For the unsupervised and partially supervised cases, the lack of identifiability is readily apparent in that we can swap various factors and factor weights tied to unsupervised states within our REDCLIFF(-S) architecture and arrive at functionally identical models.

The fully supervised case is also not identifiable, as we illustrate via a counterexample. Consider the system with $n_k = 2$ states (0 and 1) and $n_c = 2$ nodes (A and B) where both nodes are self-connected. Assume all innovations in A and B follow a symmetric random distribution (e.g., the uniform or normal distributions) in both states 0 and 1 at each time step. Finally, assume the only difference between states 0 and 1 is that in state 0 we have that A drives B (i.e., A $\rightarrow$ B) via the ReLU (that is $\mathrm{ReLU}(x) = x\mathbf{1}_{[\max(x,0)]}(x)$) function at lag $t = 2$, whereas in state 1 we have A $\rightarrow$ B is the negative ReLU (that is $\mathrm{nReLU}(x) = x\mathbf{1}_{[\min(x,0)]}(x)$) function at lag $t = 2$. In this setting, the functional form of the A $\rightarrow$ B edge cannot be uniquely identified for either state 0 or 1 when given an infinite number of samples. This is because for any given sequence of innovations $\mathcal{S} = ([A_t, B_t])_{t=1}^T$, we have that $\exists S' = ([-A_t, B_t])_{t=1}^T$ which satisfies

$$\mathrm{ReLU}(A_t) + B_t = \mathrm{nReLU}(-A_t) + B_t$$

for any $t \in \mathbb{N}_{\leq T}$. In other words, for every instance in which it appears that A is influencing B via the ReLU update rule, we will observe another instance in which A is possibly influencing B via the nReLU update *with the exact same effect*. Thus, the ReLU and nReLU update rules are equally likely in both state 0 and in state 1, rendering the two states unidentifiable by definition.

In summary, we have demonstrated that systems featuring nonlinear, dynamic causal graphs are not identifiable unless one makes sweeping assumptions, which preclude even fairly simple systems (such as our 2-node system with ReLU-like nonlinearities) from being considered. Despite this, previous work has shown that causal discovery methods still benefit scientific research in real-world settings where nonlinear, dynamic causal graphs are common (Mague et al., 2022).

## A.2. Noisy Classification vs Generative Task Information Requirements

We now provide a theoretical argument as to why it may be necessary to provide additional system history to the REDCLIFF(-S) state model $g_\theta$, and even to drastically limit its modeling capabilities (e.g. via the invertible $g_\alpha$ and $g_y$ functions of Section 3.5). Our argument centers on the implications of the parameters $\tau_{\mathrm{in}}$ and $\tau_{\mathrm{cl}}$ (Eq. 4), and is thus related to long-standing prior work investigating the effects of similar parameters on state-space models of dynamical systems (Kugiumtzis, 1996). Here, we argue that the task of predicting the state of a signal requires more historical time steps than is required to generate the same time series in the presence of nontrivial noise, at least for some dynamical systems. We now present a summary of our theoretical argument, followed by a proof sketch, and finally a discussion of the theoretical implications.

### A.2.1. MOTIVATING EXAMPLE (LEMMA 1 PRELIMINARIES)

As a motivating example, we explore a system for which a classifier needs more historical information as opposed to a generative model. In our example, there are at least two system states governed by different causal relationships operating on similar time scales, for which a classifier cannot both *a)* distinguish between these two states and *b)* be unbiased.

Let $\Phi$ be a system consisting of $D \in \mathbb{N}$ interacting variables according to $N \in \mathbb{N}$ states, in which the dynamics of $\Phi$ are governed by $M \in \mathbb{N}$ distinct causal graphs between variables. Define $\tau$ as the maximal number of historical time steps required to generate the behavior of $\Phi$ across all states[6]. Assume $\exists$ two states of $\Phi$ (call them states $i$ and $j$ where $i \neq j$) for

---

[6]Effectively, $\tau$ is the number of time steps required to reduce $\Phi$ to a Markov Chain.

which the generative functions can be expressed as $f_i(\mathbf{x}_t) = \mathbf{A}_i \mathbf{x}_{t-\tau}$ and $f_j(\mathbf{x}_t) = \mathbf{A}_j \mathbf{x}_{t-\tau}$ where both $\mathbf{A}_i \in \mathbb{R}^{D \times D}$ and $\mathbf{A}_j \in \mathbb{R}^{D \times D}$ are invertible with $\mathbf{A}_i \neq \mathbf{A}_j$.

Finally, let $\mathbf{X}_i \in \mathbb{R}^{D \times T}$ be a recorded signal of $T \in \mathbb{N}$ observations made while $\Phi$ was in state $i \leq N$. We say that $\mathbf{X}_i$ exhibits nontrivial noise if $\exists$ another recording $\mathbf{X}_j \in \mathbb{R}^{D \times T}$ such that $0 < j \leq N$ with $j \neq i$ for which the initial conditions are independently, identically distributed with those of $\mathbf{X}_i$, that is: $\mathbf{X}_{i_{:,0}} \overset{\text{i.i.d.}}{\sim} \mathbf{X}_{j_{:,0}}$. [7]

### A.2.2. LEMMA 1: STATEMENT AND PROOF SKETCH

**Lemma 1** To learn an accurate and unbiased state model $g_\theta$ mapping recorded signals with nontrivial noise to states of $\Phi$ (i.e., $g_\theta : \mathbb{R}^{D \times T} \to \mathbb{N}_{\leq N}$), it must be true that $T > \tau$.

**Proof Sketch** Lemma 1 follows from Bayes' Theorem, which we use to show that two windows of length $\tau$ sampled from a distribution featuring nontrivial noise cannot be readily distinguished by an unbiased model $g_\theta$.

Suppose the recording $\mathbf{x} \in \mathbb{R}^{D \times T}$ with $T < \tau$ is randomly sampled from $\Phi$ according to a uniform distribution over all $N$ states. By way of contradiction, assume we have an unbiased classifier (i.e., the REDCLIFF-S state model $g_\theta$) for which $p(f_i|\mathbf{X}) \neq p(f_j|\mathbf{X})$.

Notice that since both $\mathbf{A}_i$ and $\mathbf{A}_j$ are invertible, we have both that $\mathbf{X}_{:,T-\tau} = f_i^{-1}(\mathbf{X}_{:,T}) = \mathbf{A}_i^{-1} \mathbf{X}_{:,T} \sim p(\mathbf{X}|f_i)$ as well as that $\mathbf{X}_{:,T-\tau} = f_j^{-1}(\mathbf{X}_{:,T}) = \mathbf{A}_j^{-1} \mathbf{X}_{:,T} \sim p(\mathbf{X}|f_j)$.

Hence, we have

$$\implies p(\mathbf{X}|f_i) = p(\mathbf{X}|f_j). \tag{12}$$

Applying Bayes' Theorem, we have that

$$\implies \frac{p(f_i, \mathbf{X})p(\mathbf{X})}{p(f_i)} = p(\mathbf{X}|f_i) = p(\mathbf{X}|f_j) = \frac{p(f_j, \mathbf{X})p(\mathbf{X})}{p(f_j)} \implies \frac{p(f_i, \mathbf{X})}{p(f_i)} = \frac{p(f_j, \mathbf{X})}{p(f_j)} \tag{13}$$

Now recall that $\mathbf{X}$ was randomly sampled according to a uniform distribution over states. Therefore, $p(f_i, \mathbf{X}) = p(f_j, \mathbf{X})$, which yields

$$\implies \frac{1}{p(f_i)} = \frac{1}{p(f_j)} \implies p(f_i) = p(f_j) \tag{14}$$

However, we can also apply Bayes' Theorem to the output of our classifier $p(f_i|\mathbf{X}) \neq p(f_j|\mathbf{X})$. Doing this yields

$$\frac{p(f_i, \mathbf{X})p(f_i)}{p(\mathbf{X})} = p(f_i|\mathbf{X}) \neq p(f_j|\mathbf{X}) = \frac{p(f_j, \mathbf{X})p(f_j)}{p(\mathbf{X})} \tag{15}$$

As before, we use the fact that $p(f_i, \mathbf{X}) = p(f_j, \mathbf{X})$ to obtain

$$\implies p(f_i, \mathbf{X})p(f_i) \neq p(f_j, \mathbf{X})p(f_j) \implies p(f_i) \neq p(f_j) \tag{16}$$

But the finding that $p(f_i) \neq p(f_j)$ contradicts our assumption that our classifier was unbiased, for which $p(f_i) = p(f_j)$.

$\Rightarrow\!\Leftarrow$

### A.2.3. DISCUSSION

Lemma 1 has important implications for training REDCLIFF-based models. **Effectively, once the state model $g_\theta$ in a REDCLIFF model has enough capacity to accurately distinguish between states, Lemma 1 indicates that $g_\theta$ may also have access (by necessity) to at least as much information as that required to forecast the evolution of $\Phi$ (assuming the proper noise profile) in one or more states of the system**, at least for restricted (yet simple and arguably common) classes of temporal systems. These findings indicate that it may be necessary to restrict the modeling capabilities of $g_\theta$ to ensure the proper delineation of tasks within a REDCLIFF-S model.

---

[7]This is relevant to hypothesis generation, where we often start by assuming signals are generated from similarly distributed initial conditions, and we attempt to identify where this assumption fails.

### A.3. 'Broadcast Multiplication': Formal Definition

Here we formally define broadcast multiplication, the linear operation used to combine outputs of our generative factors.

**Definition**: Let $N$-dimensional vectors $\mathbf{v}, \mathbf{v}' \in \mathbb{R}^N$ be given along with 3-axis tensors $\mathbf{Z}, \mathbf{Z}' \in \mathbb{R}^{N \times P \times Q}$ where both $P, Q \in \mathbb{N}$. Then 'broadcast multiplication', denoted by $\odot^\dagger$, is an operation such that

$$\mathbf{v} \odot^\dagger \mathbf{Z} = \sum_{n=1}^{N} \mathbf{v}_n \cdot \mathbf{Z}_{n,:,:} \tag{17}$$

## B. Methodological Details

### B.1. Factor-level Implementation - A Brief Review of cMLPs (Tank et al., 2021)

As mentioned in the main paper, we implement each factor of the REDCLIFF-S models in this work as a "component-wise Multi-Layer Perceptron" or "cMLP" (Tank et al., 2021); here we briefly summarize the cMLP architecture and refer the reader to the original paper for more details. Essentially, the cMLP architecture is a 1D convolutional neural network that treats the first layer of weights as an information filter mapping prior variable values of a dynamic system to the next predicted value of a single variable. Since the architecture allows for a set of historical system observations to be input, the first layer of a cMLP comes to represent a set of estimated, time-lagged Granger causal connections. One can view the magnitude of each weight in the first layer of parameters as the strength of a Granger causal connection from one of the input variables to the predicted variable. By instantiating a different cMLP for each variable in a dynamical system, one can learn a sequence of time-lagged, full adjacency matrices mapping prior variable values to future variable values. We found this architecture to be flexible for our purposes and it proved at least as effective in predictive tasks as other architectures we tested in preliminary experiments (namely the cLSTM (Tank et al., 2021) and DGCNN (Song et al., 2018) architectures).

### B.2. Standardizing Causal Graph Estimates

To facilitate fair comparison between both baseline and our REDCLIFF-based algorithms, we needed to standardize the form in which each algorithm's estimated Granger causal graphs were presented. This was nontrivial, since some of the algorithms incorporated a lag-dimension into their causal graph estimates (esp. those based on the cMLP architecture from Tank et al. (2021)), while others (esp. DYNOTEARS from Pamfil et al. (2020)) treated each feature measured from a channel - be it a historical scalar value or a power spectral feature - as an individual node in the estimated graph, while yet others (esp. dCSFA-NMF in Gallagher et al. (2021) and REDCLIFF-based models) learned multiple causal graph factors.

The standard we chose to adopt was that each estimated (and true) Granger causal graph would need to be presented as a simple adjacency matrix; that is, if there were $n_c$ variables in the recorded system, then the true/estimated graph(s) from each system/model were compressed to lie within $\mathbb{R}^{n_c \times n_c}$. In practice, this meant that graphs involving lags and/or node features had their representations compressed by summing over the lag and/or feature values corresponding to each node.

If an algorithm could only estimate a single graph for a system governed by $n_k > 1$ graphs, the estimated graph was copied $n_k$ times and compared to each of the $m$ true causal graphs, with results averaged across each comparison.

### B.3. REDCLIFF-S Stopping Criteria and Model Selection

Here we describe how we arrived at the stopping criteria for REDCLIFF-S model training and how it was used in model selection. Our stopping criteria are summarized in Equation 10, which has three main components: a forecasting term, a state classification term, and an estimated Granger causal graph similarity term. Intuitively, the two MSE terms (forecasting and state classification) are in the stopping criteria to ensure that our performance on the two main subtasks of our REDCLIFF-S models (predicting factor effect and relevance; see Figure 1A) is being optimized.

The third cosine similarity-based term (estimated Granger causal graph similarity) is included for several reasons. For one thing, the very premise of REDCLIFF-S is that the target system being modeled has *different* causal relationships over time; hence, our third term implicitly assumes that factors should be different by minimizing this estimated Granger causal graph similarity term. A potentially useful side effect of minimizing similarity between factors is that it reduces the size and complexity of our REDCLIFF-S model.

In addition to these intuitive motivations for minimizing cosine similarity between factors, we also found this cosine

similarity-based term was correlated with the value of various graph similarity metrics comparing factor estimates with their ground truth representations over the course of training (for example, see Supplemental Table 4). Other interesting phenomena include our finding that troughs in our mean cosine similarity curve coincided with peaks in the DeltaCon0 similarity (a measure of functional similarity between two graphs (Koutra, 2015)) between estimated and true causal graphs, as well as the onset of instabilities in the ROC-AUC curves when comparing estimates to ground truth. These observations (along with similar ones from many other experiments) demonstrate why this penalty for the sum of cosine similarity between factor estimates served as a reasonable proxy to determine when to stop training.

*Table 4.* Comparing correlation between stopping criteria elements and the ROC-AUC between final REDCLIFF-S factor predictions of causal relationships and the true factor graphs in our Synthetic Systems grid search over REDCLIFF-S model parameters. Values rounded to three significant figures, with the strongest negative correlation bolded.

| LOSS/PENALTY TERMS INCLUDED IN CRITERIA | | | |
| FORECASTING | STATE PREDICTION | GRAPH SIMILARITY | PEARSON'S R VALUE |
|:---:|:---:|:---:|:---:|
| ✓ | - | - | -0.368 |
| - | ✓ | - | 0.529 |
| - | - | ✓ | -0.332 |
| ✓ | ✓ | - | -0.176 |
| ✓ | - | ✓ | -0.654 |
| - | ✓ | ✓ | -0.201 |
| ✓ | ✓ | ✓ | **-0.674** |

One open question is how to balance the significant differences in variation between the forecasting and state prediction penalties on the one hand and the significantly less-variable cosine similarity term on the other. In selecting the final model for our D4IC experiment, we found that weighting the cosine similarity penalty by $60\rho$ (instead of just $\rho$) in our selection criteria balanced the three variation scales to all roughly lie within the same order of magnitude. More sophisticated balancing schemes would likely improve results.

### B.4. Discussion: Limitations and Future Work

Most limitations of REDCLIFF-S arise as trade-offs for scientific utility. For example, the set of REDCLIFF-S factors has many hyperparameters that can be costly to tune; this cost restricted us to tuning on a single repeat of our 12-node, 33-edge, 3-factor system in Section 4.2, and may have limited REDCLIFF-S' measured performance.

We also wish to draw attention to a general lack of methods for measuring functional similarity in nonlinear graphs. As discussed in relation to Equation 5, we found it helpful to approximate the DeltaCon0 metric (which measures functional/compositional similarity between lagged linear graphs) (Koutra, 2015) using a cosine similarity penalty; however, to the best of our knowledge, the nonlinear version of DeltaCon0 has not been developed. This poses a major obstacle to evaluating the performance of models on our Synthetic Systems task, and is a research question we hope our work motivates.

### B.5. Comparing State Prediction Error with 1-Factor Models

To test the ability of the REDCLIFF-S algorithm to dynamically adjust factor weightings at each time step in preliminary experiments, we compared the mean squared error of each state label prediction $\hat{\mathbf{y}}$ made by REDCLIFF-S models against a 'Naïve 1-Factor Model' state prediction. For this, we adopted the convention that a state label prediction made by a 1-factor model (such as the cMLP and cLSTM models by (Tank et al., 2021) or the DGCNN method by (Song et al., 2018)) would be equivalent to predicting that all system factors (represented by the model's single factor) would always be present; hence, $\hat{\mathbf{y}} = \mathbf{1}$ for any input $\mathbf{x}_{in}$. The difference in MSE between these naïve predictions and the true label $\mathbf{y}$ was taken with the MSE obtained by the predictions of the REDCLIFF-S models across many samples and then averaged.

## C. Additional Experimental Results

### C.1. Pairwise Optimal F1-Score Improvement of REDCLIFF-S on Synthetic Systems' 6-2-2 Data

As we alluded to in the abstract of the main paper, REDCLIFF-S "improves f1-scores of predicted dynamic causal patterns by roughly 22-28% on average over baselines in some of our experiments, with some improvements reaching well over

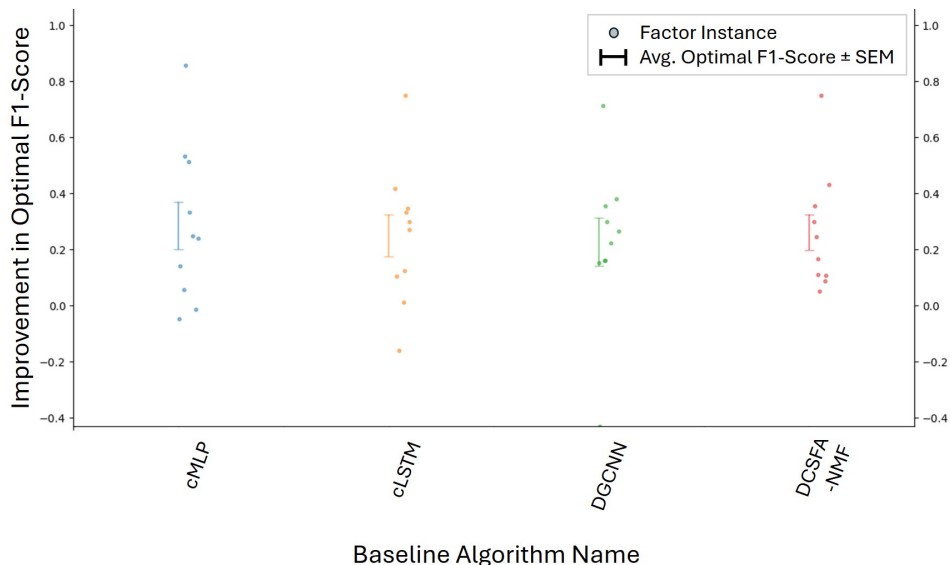

*Figure 6.* Pairwise optimal f1-Score Improvement of REDCLIFF-S on the Synthetic Systems 6-2-2 experiment.

60%"; Figure 6 plots these levels of improvement explicitly.

## C.2. Additional D4IC Results

We report the main results of our D4IC Low Signal-to-Noise Ratio (LSNR) experiments in Figure 7 and those of our D4IC Moderate Signal-to-Noise Ratio (MSNR) experiments in Figure 8.

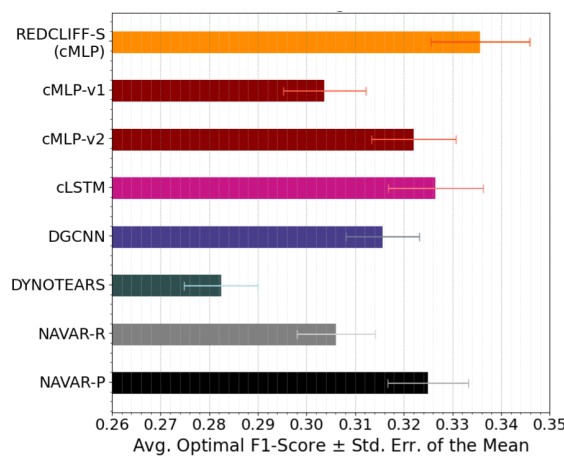

*Figure 7.* Optimal f1-score on D4IC-LSNR experiments.

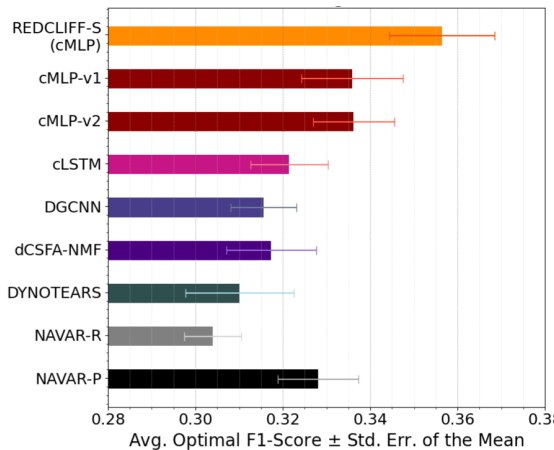

*Figure 8.* Optimal f1-score on D4IC-MSNR experiments.

We also report ROC-AUC statistics corresponding to REDCLIFF-S, cMLP-v2, and dCSFA-NMF models trained on the D4IC dataset in Table 5.

## C.3. Additional Synthetic Systems Results

We now report additional results from our Synthetic Systems experiments for transparency.

*Table 5.* Average inter-variable ROC-AUC score $\pm$ SEM on the D4IC LSNR, MSNR, and HSNR datasets. Averages are taken across factors and repeats. Note that predictions included edge weighting, but we converted true-edge weights into binary 'present'/'not-present' values during evaluation to facilitate ROC-AUC calculations.

| | D4IC NOISE LEVEL | | |
| ALGORITHM | LSNR | MSNR | HSNR |
| --- | --- | --- | --- |
| CMLP-V2 | 0.550$\pm$0.016 | 0.594$\pm$0.017 | 0.567$\pm$0.015 |
| DCSFA-NMF | - | 0.567$\pm$0.013 | 0.554$\pm$0.013 |
| **REDCLIFF-S (CMLP)** | **0.632$\pm$0.014** | **0.635$\pm$0.016** | **0.629$\pm$0.018** |

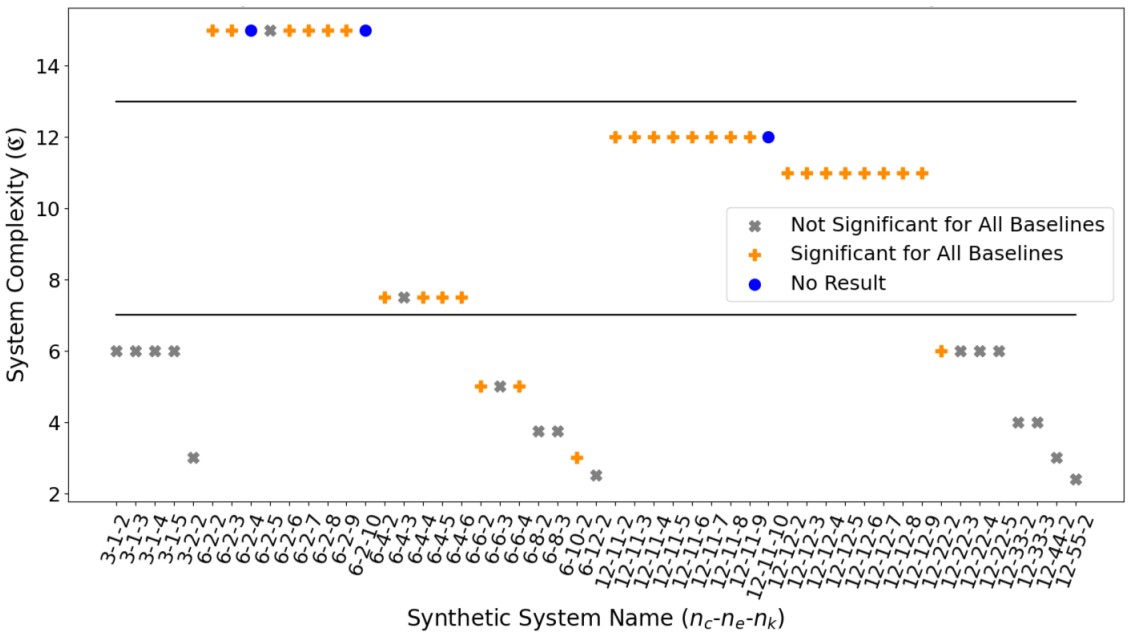

*Figure 9.* Summary plot of pairwise optimal f1-score improvement of REDCLIFF-S across Synthetic Systems experiments.

In Table 6 we report ROC-AUC scores corresponding to the models and optimal f1-score statistics in Figure 2 of the main manuscript.

We give additional results related to Table 2 in Supplemental Table 7. Note that between the two tables, the top performing baselines - usually Regime-PCMCI (Saggioro et al., 2020) and SLARAC (Weichwald et al., 2020) - appear to 'trade off' between performance on metrics such as optimal f1-score and structural hamming distance (SHD) versus performance on the AID metrics. In contrast, REDCLIFF-S does not exhibit this trade-off, regularly attaining competitive performance (statistically similar to the best or second-best performance) across all metrics.

Regarding optimal f1-score performance, we provide a summary plot of these results in Supplemental Figure 9 and a visualization comparing true to estimated factors in Supplemental Figure 10. Optimal f1-score and ROC-AUC score results are grouped by systems rated with similar complexity scores. Scores of $\mathfrak{C} \leq 7.0$ received a 'Low' categorization; results on systems with this rating can be found in Supplementary Figures 11 through 16. Scores that satisfied $7.0 < \mathfrak{C} \leq 13.0$ were categorized as 'Moderate' complexity; results on systems with this rating can be found in Supplementary Figures 17 through 22. Scores were categorized as 'High' complexity if they satisfied $\mathfrak{C} > 13.0$; the results on systems with this rating can be found in Supplementary Figures 23 through 24.

*Table 6.* Average inter-variable ROC-AUC score $\pm$ SEM on the Synthetic System 6-2-2, 6-4-2, 12-11-2, and 12-11-5 datasets. Averages are taken across factors and repeats. Note that predictions included edge weighting, but we converted true-edge weights into binary 'present'/'not-present' values during evaluation to facilitate ROC-AUC calculations.

| | SYNTHETIC SYSTEM | | | |
| ALGORITHM | 6-2-2 | 6-4-2 | 12-11-2 | 12-11-5 |
| --- | --- | --- | --- | --- |
| cMLP | 0.544±0.035 | 0.591±0.021 | 0.572±0.020 | 0.562±0.015 |
| cLSTM | 0.612±0.050 | 0.534±0.040 | 0.532±0.024 | 0.552±0.020 |
| DGCNN | 0.556±0.072 | 0.566±0.036 | 0.562±0.030 | 0.536±0.015 |
| dCSFA-NMF | - | - | - | 0.557±0.012 |
| **REDCLIFF-S (cMLP)** | **0.797±0.036** | **0.682±0.037** | **0.756±0.019** | **0.743±0.019** |

*Table 7.* Supplemental results to Table 2. Note that the final column now gives the average placement across metrics from both tables combined. Again, note that performance is averaged across 10 random seeds, leading to slight differences with REDCLIFF-S performance given in Table 6.

| | (NO THRESHOLD) | ANCESTOR AID ERROR | | OSET AID ERROR | | OVERALL |
| ALGORITHM | ROC-AUC | UPPER | LOWER | UPPER | LOWER | AVG. PLACE. |
| --- | --- | --- | --- | --- | --- | --- |
| LASAR (SUP.) | 0.457±0.043 | 4.9±1.466 | 2.9±0.434 | 4.7±1.331 | 2.9±0.434 | 3.8 |
| QRBS (SUP.) | 0.424±0.029 | 2.1±0.385 | 1.9±0.537 | 2.5±0.548 | 1.9±0.537 | 3.0 |
| SLARAC (SUP.) | 0.347±0.023 | 0.8±0.716 | 0.4±0.219 | 0.7±0.626 | 0.4±0.219 | 2.8 |
| REGIME-PCMCI | 0.775±0.018 | 3.8±0.540 | 3.6±0.518 | 4.1±0.477 | 3.8±0.502 | 3.1 |
| **REDCLIFF-S** | 0.782±0.021 | 2.2±0.335 | 2.1±0.434 | 2.7±0.390 | 2.3±0.522 | **2.3** |

## C.4. Additional Region-Averaged TST 100 Hz Results

We provide a visualization of our approach to model selection (specifically with respect to determining the number of REDCLIFF-S factors) in Supplemental Figure 25. Plots of the supervised adjacency matrices learned by our REDCLIFF-S models trained on the Region-Averaged TST 100 Hz dataset can be found in Supplemental Figures 26 through 28. We provide additional plots of the differences between the average Open Field (OF) and Tail Suspended (TS) supervised factors in Supplemental Figure 29 and between the average TS and Home Cage (HC) supervised factors in Supplemental Figure 30.

## C.5. Region-Averaged Social Preference 100 Hz Results

In addition to the TST dataset, we also applied our REDCLIFF-S method to analyze the Social Preference (SP) dataset described in Mague et al. (2022). It should be noted that our analysis here is preliminary and was not as thorough as our analysis of the TST data due to time constraints. Supplemental Figure 31 provides a visualization of how we selected the number of factors for the REDCLIFF-S models trained on different folds of the SP data; note that the selection criteria history does not decrease monotonically as the number of factors increases (as it did on the TST data), suggesting that a more detailed search of REDCLIFF-S' hyperparameter settings may yield more accurate results. Plots of the supervised adjacency matrices learned by our REDCLIFF-S models trained on the Region-Averaged SP 100 Hz dataset can be found in Supplemental Figures 32 and 33.

We provide an additional plot of the mean normalized difference between Social Preference (SP) and Object Preference (OP) supervised factors in Supplemental Figure 34. While several of the connections found by REDCLIFF-S do appear to have been published previously (compare with Figure 4A in Mague et al. (2022)), we highlight the fact that the network identified by REDCLIFF-S seems to break from previously published results by including the Hippocampus in multiple network edges. Interestingly, Mague et al. (2022) report that elevated power in certain frequency bands recorded in the Hippocampus was implicated in social behavior, but they did not seem to detect any coherence between activity in the Hippocampus and other regions. Given that our REDCLIFF-S method is designed to estimate both linear and nonlinear granger causal relationships, whereas the dCSFA-based technique employed by Mague et al. (2022) focuses exclusively on linear causal relationships, these findings may suggest that the Hippocampus shares nonlinear causal relationships with other regions of the brain during social activity. More follow-up work would need to be done to confirm this hypothesis, but we are excited to see that REDCLIFF-S appears to be identifying plausible, novel hypotheses.

*Table 8.* Mean optimal f1-score $\pm$ SEM obtained by REDCLIFF-S ablations across various datasets (summarized in Table 3 of the main paper). Values should be compared against those depicted in Figures 2A and 4 of the main paper.

| | ABLATION | | | |
|---|---|---|---|---|
| DATASET | $\rho = 0$ (EQ. 5, 10) | $n_k = 1$ ($\equiv$ EQ. 2) | $\alpha = 1$ (EQ. 4) | $\lambda = 0$ (EQ. 9) |
| SYNTHETIC SYSTEM 6-2-2 | 0.463$\pm$0.057 | 0.279$\pm$0.044 | 0.291$\pm$0.045 | 0.279$\pm$0.044 |
| SYNTHETIC SYSTEM 6-4-2 | 0.517$\pm$0.054 | 0.335$\pm$0.031 | 0.407$\pm$0.037 | 0.335$\pm$0.031 |
| SYNTHETIC SYSTEM 12-11-2 | 0.296$\pm$0.037 | 0.188$\pm$0.033 | 0.208$\pm$0.018 | 0.188$\pm$0.033 |
| SYNTHETIC SYSTEM 12-11-5 | 0.362$\pm$0.031 | 0.156$\pm$0.014 | 0.197$\pm$0.018 | 0.156$\pm$0.015 |
| D4IC HSNR | 0.370$\pm$0.014 | 0.304$\pm$0.002 | 0.336$\pm$0.009 | 0.342$\pm$0.010 |

## C.6. Ablation Analyses

In Table 8 we provide details of our ablation experiments summarized in Table 3 of the main paper. With the exception of our $\rho = 0$ ablation experiments, we see that the ablation scores are worse - in many cases by a large margin - than those of the full REDCLIFF-S model, suggesting that these hyperparameters play an important role in the performance of REDCLIFF-S.

The $\rho = 0$ ablation experiments tell a more nuanced story, with performance degrading marginally for three of the five systems tested and actually increasing when $\rho$ was ablated for two of the systems. These observations are somewhat unsurprising for a few reasons. Firstly, we reiterate that we only tuned the hyperparameters - including $\rho$ - on a single synthetic system configuration due to time constraints, so it is unsurprising that at least one of our parameter settings is suboptimal in certain cases. Secondly, we note that there appears to be a pattern in Table 8 where the systems with relatively fewer edges and factors (esp. the 6-2-2 and 12-11-2 Synthetic Systems) tend to perform worse after setting $\rho = 0$, whereas those with more edges and factors (esp. the 6-4-2 and 12-11-5 Synthetic Systems) see improvements when $\rho = 0$. This makes intuitive sense, since state graphs in systems with more edges and factors are simply more likely to share similar causal relationships than those with fewer connections and states. Finally, the CosSim loss term modulated by $\rho$ was introduced into the REDCLIFF training process after we observed that doing so was loosely correlated with improved ROC-AUC performance at test time in preliminary experiments (see Appendix B.3); since the ROC-AUC and F1 metrics are fundamentally different, the correlated relationship between minimized CosSim and ROC-AUC may not always indicate improved F1 performance. In summary, it would seem that $\rho$ is more sensitive than other hyperparameters to properties of the system being modeled and should thus be carefully tuned to suit the particular target system (and possibly dropped entirely for sufficiently dense or expressive systems).

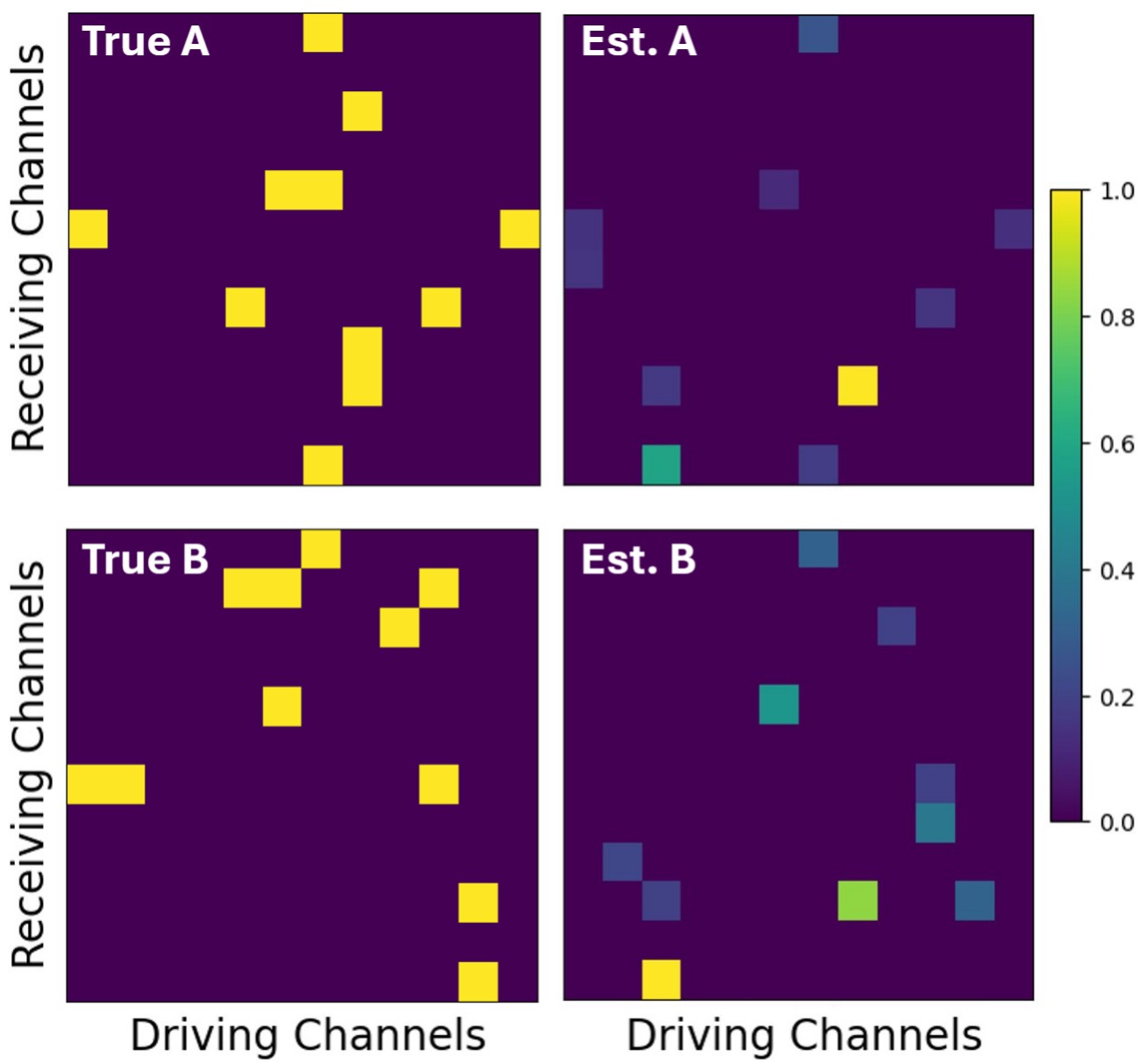

*Figure 10.* Visualizing true (left column) and REDCLIFF-S Top-10 estimated (right column) inter-variable causal relationships of two factors ('A' - also depicted in main paper Figure 3 - and 'B') from Synthetic System "12-11-5". The color map shows (normalized) weight of inter-variable causal relationships.

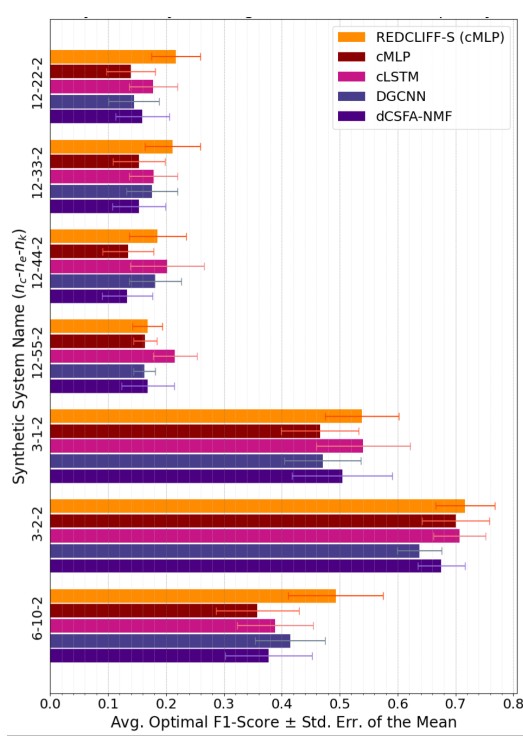

Figure 11. Average optimal f1-scores $\pm$ SEM from our Low-complexity Synthetic Systems experiments.

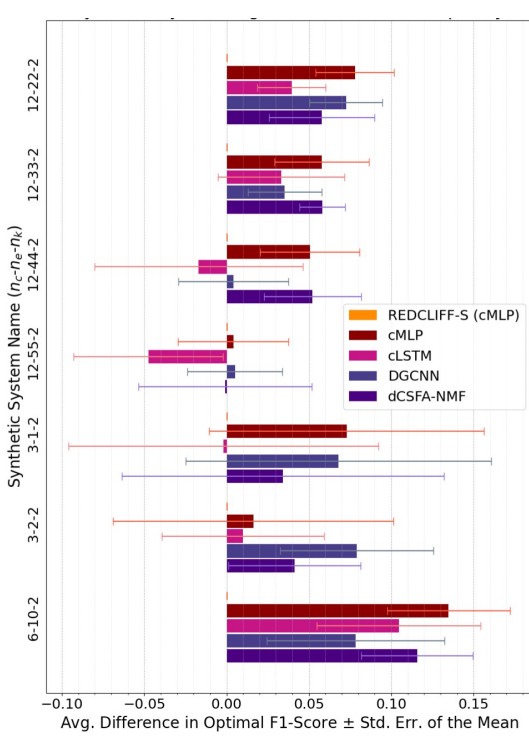

Figure 12. Average improvement in optimal f1-scores $\pm$ SEM by the REDCLIFF-S algorithm over baselines from our Low-complexity Synthetic Systems experiments.

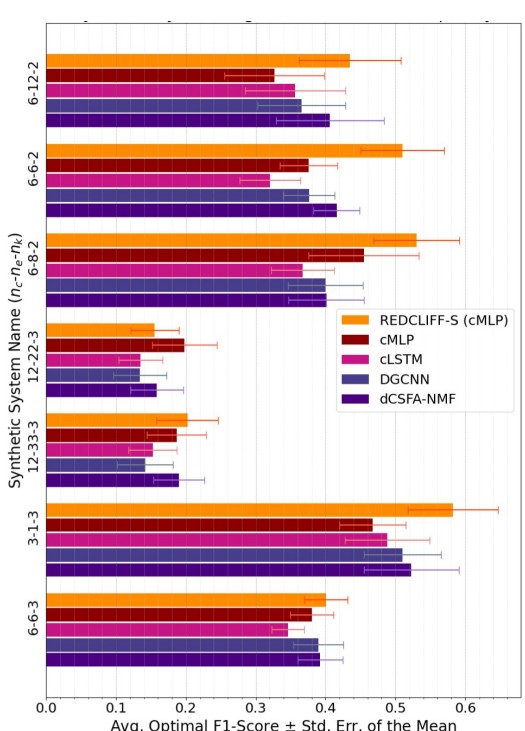

Figure 13. Average optimal f1-scores $\pm$ SEM from our Low-complexity Synthetic Systems experiments.

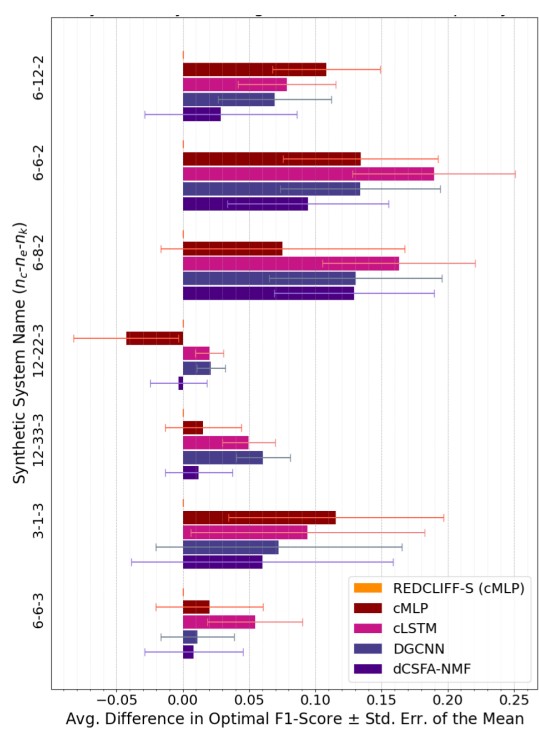

Figure 14. Average improvement in optimal f1-scores $\pm$ SEM by the REDCLIFF-S algorithm over baselines from our Low-complexity Synthetic Systems experiments.

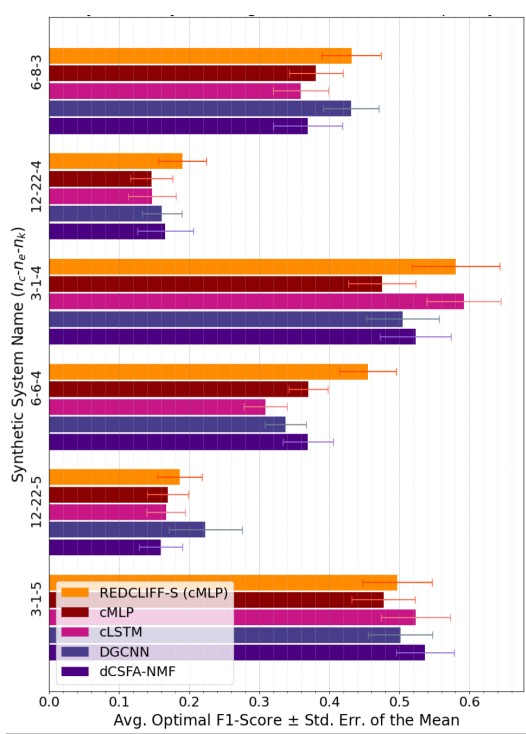

Figure 15. Average optimal f1-scores ± SEM from our Low-complexity Synthetic Systems experiments.

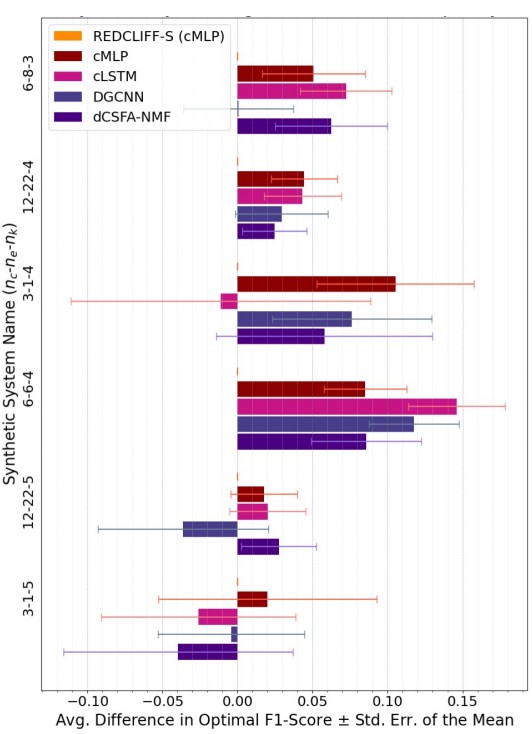

Figure 16. Average improvement in optimal f1-scores ± SEM by the REDCLIFF-S algorithm over baselines from our Low-complexity Synthetic Systems experiments.

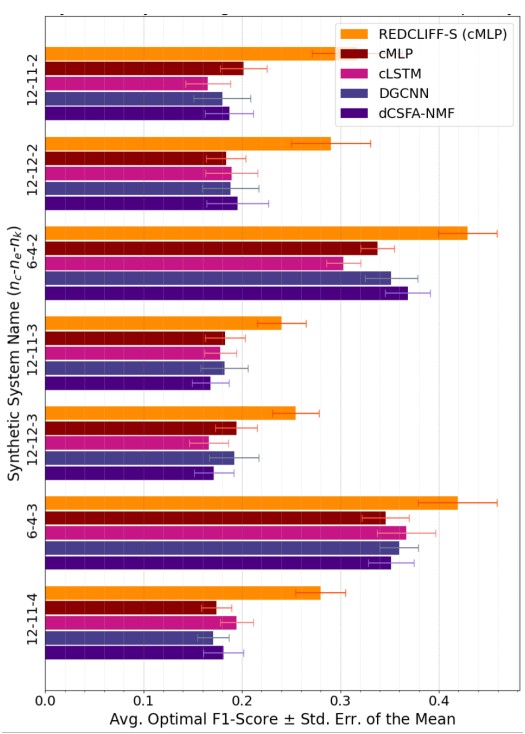

Figure 17. Average optimal f1-scores ± SEM from our Moderate-complexity Synthetic Systems experiments.

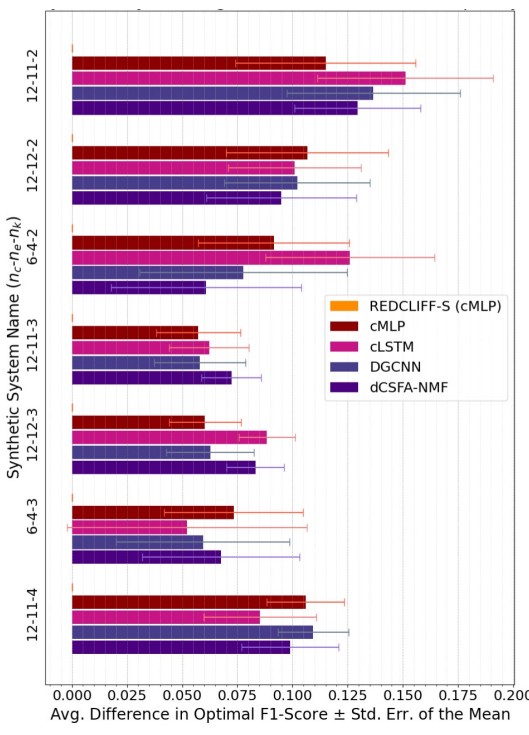

Figure 18. Average improvement in optimal f1-scores ± SEM by the REDCLIFF-S algorithm over baselines from our Moderate-complexity Synthetic Systems experiments.

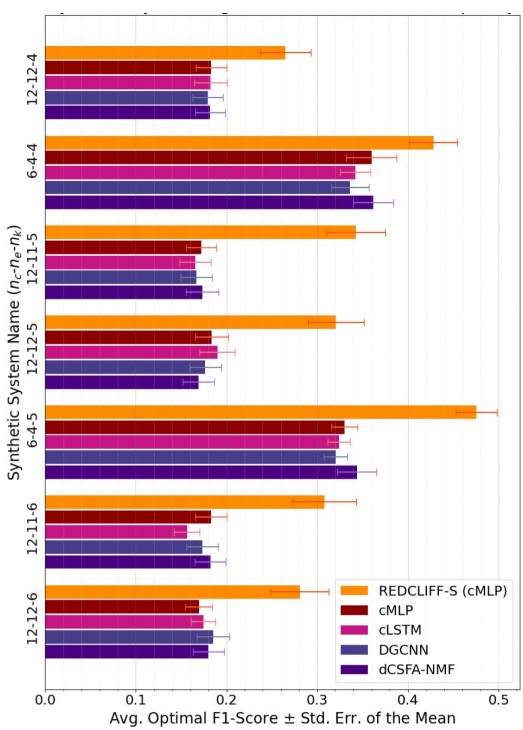

Figure 19. Average optimal f1-scores $\pm$ SEM from our Moderate-complexity Synthetic Systems experiments.

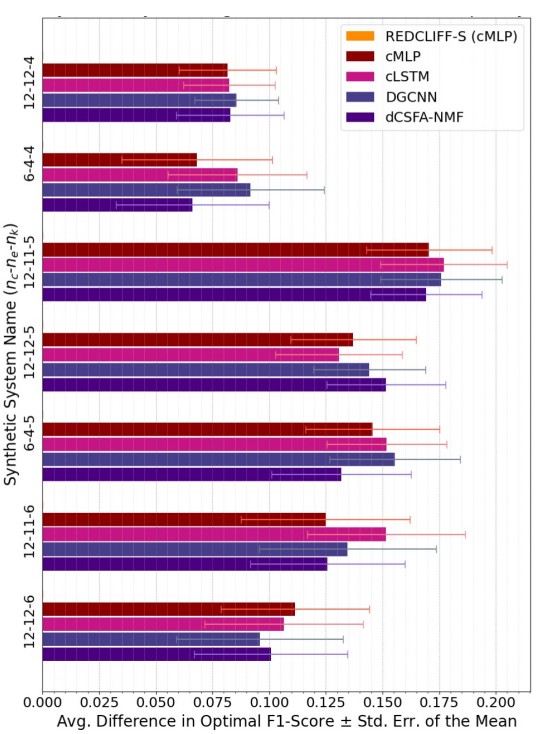

Figure 20. Average improvement in optimal f1-scores $\pm$ SEM by the REDCLIFF-S algorithm over baselines from our Moderate-complexity Synthetic Systems experiments.

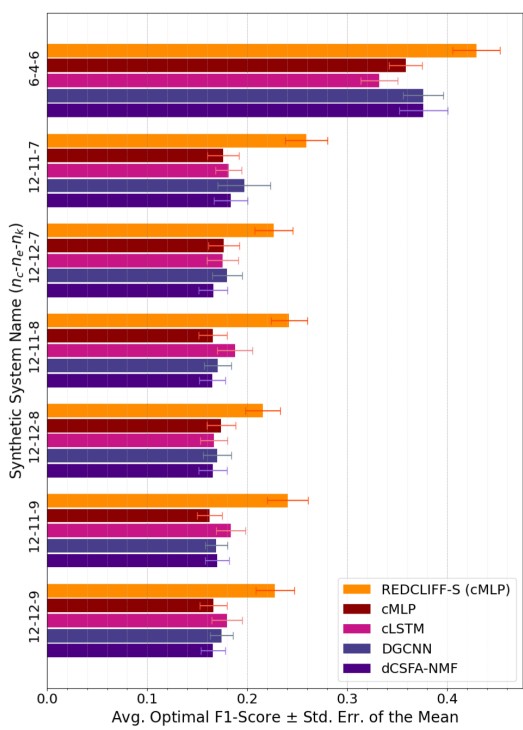

Figure 21. Average optimal f1-scores $\pm$ SEM from our Moderate-complexity Synthetic Systems experiments.

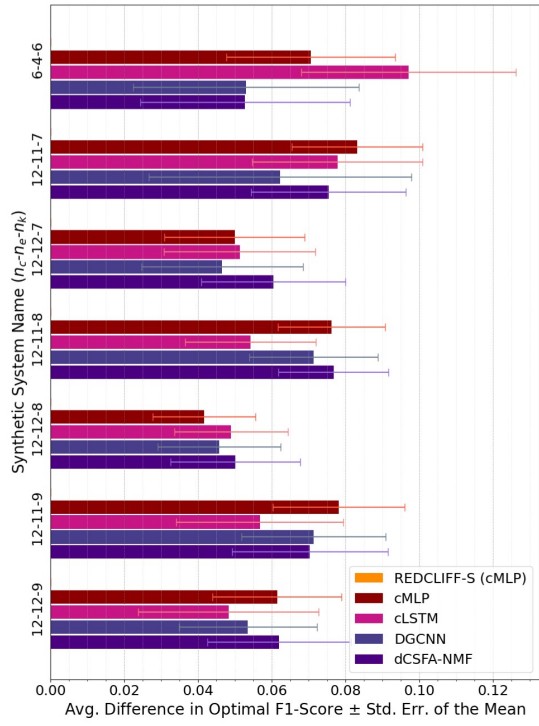

Figure 22. Average improvement in optimal f1-scores $\pm$ SEM by the REDCLIFF-S algorithm over baselines from our Moderate-complexity Synthetic Systems experiments.

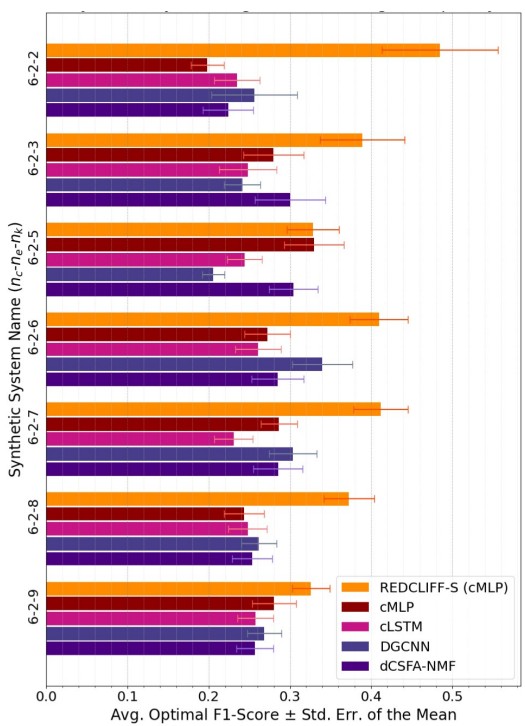

Figure 23. Average optimal f1-scores $\pm$ SEM from our High-complexity Synthetic Systems experiments.

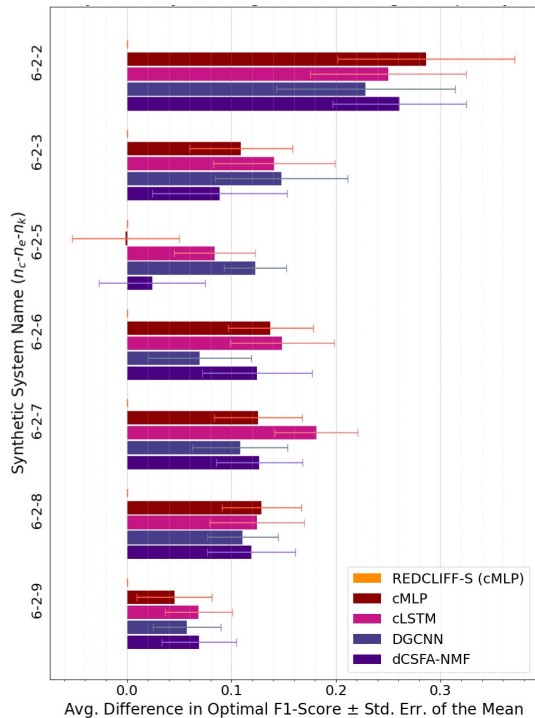

Figure 24. Average improvement in optimal f1-scores $\pm$ SEM by the REDCLIFF-S algorithm over baselines from our High-complexity Synthetic Systems experiments.

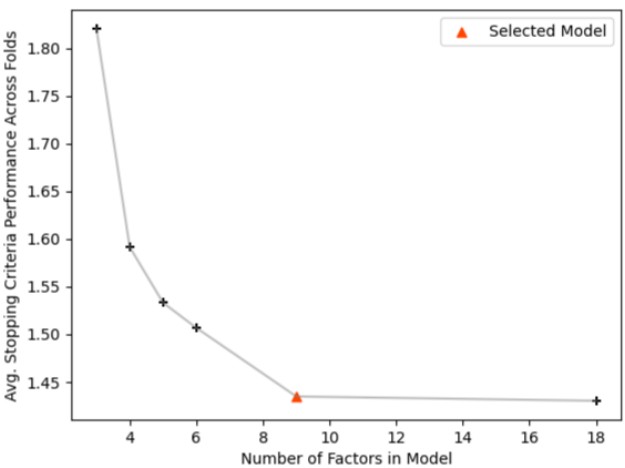

Figure 25. Visualizing model selection in the Region-Averaged TST 100 Hz case study.

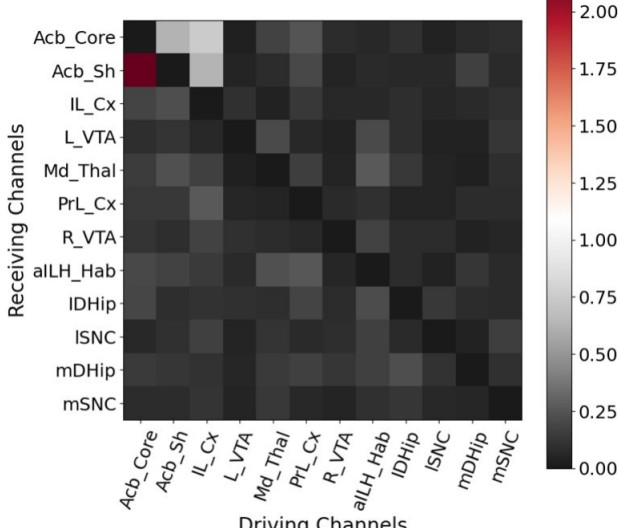

Figure 26. Average estimated strength of causal relationships between channels recorded in the Home Cage state from the Region-Averaged TST 100 Hz experiment.

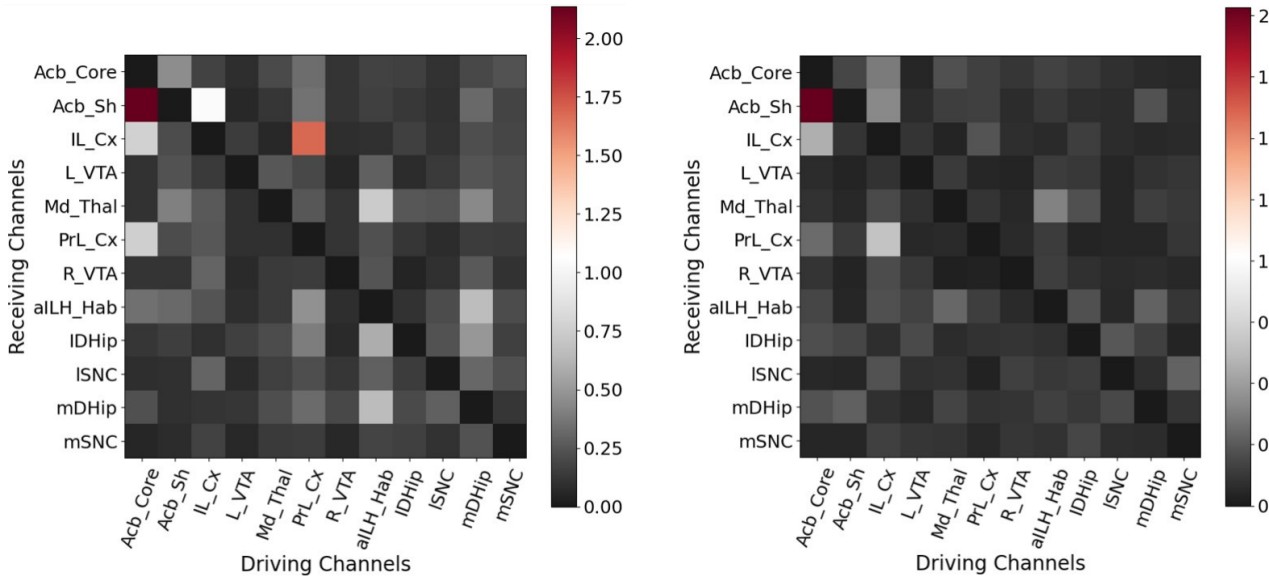

*Figure 27.* Average estimated strength of causal relationships between channels recorded in the Open Field state from the Region-Averaged TST 100 Hz experiment.

*Figure 28.* Average estimated strength of causal relationships between channels recorded in the Tail Suspended state from the Region-Averaged TST 100 Hz experiment.

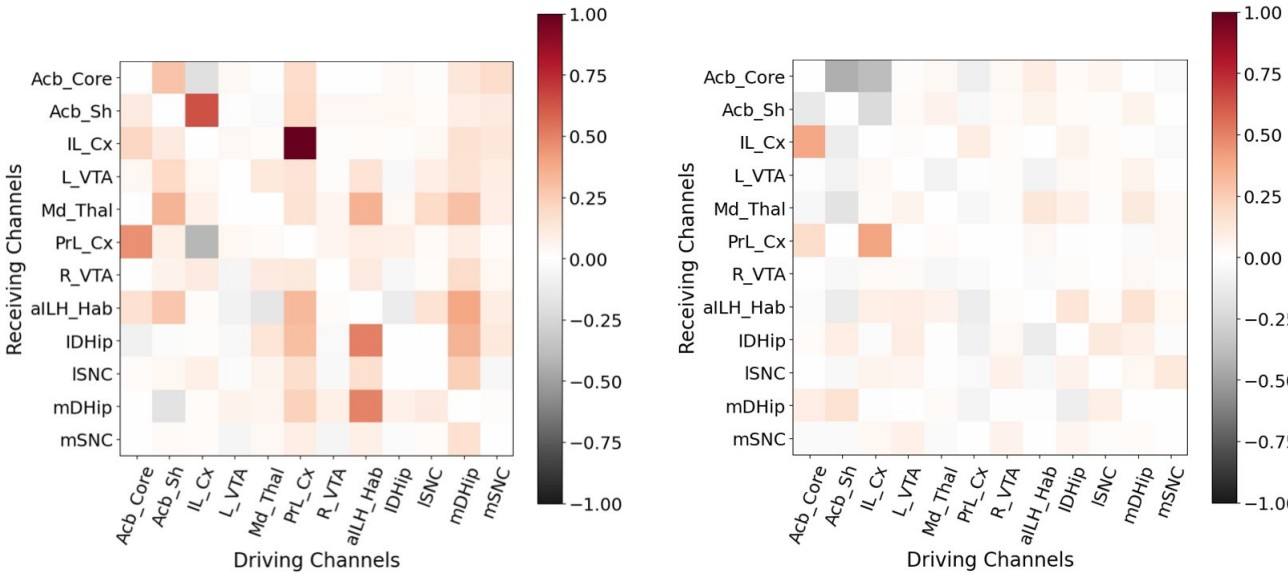

*Figure 29.* Difference between Open Field vs Tail Suspended average estimated strength of causal relationships between channels from the Region-Averaged TST 100 Hz experiment.

*Figure 30.* Difference between Tail Suspended vs Home Cage average estimated strength of causal relationships between channels from the Region-Averaged TST 100 Hz experiment.

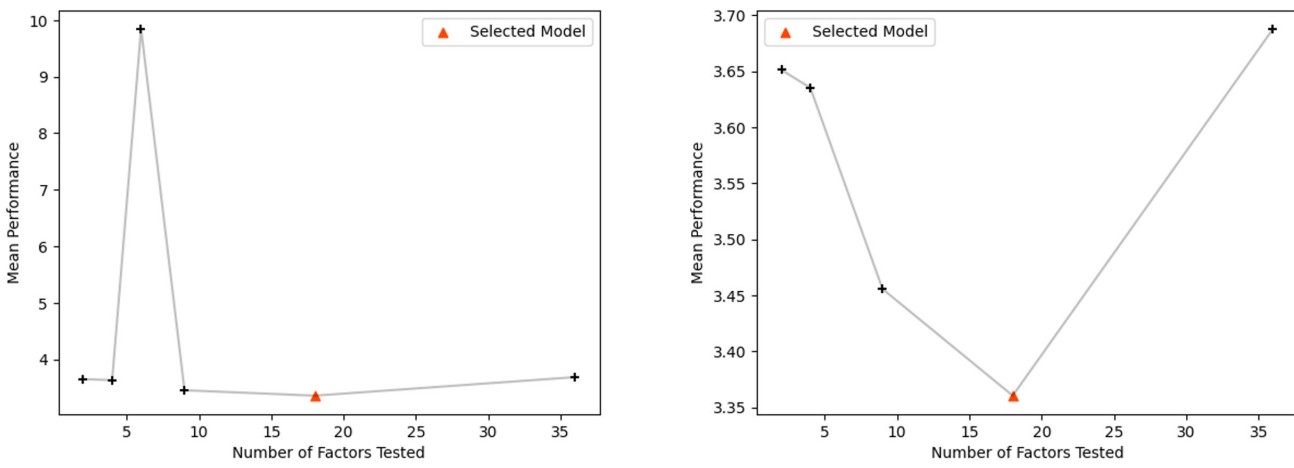

*Figure 31.* Visualizing model selection in the Region-Averaged SP 100 Hz case study.

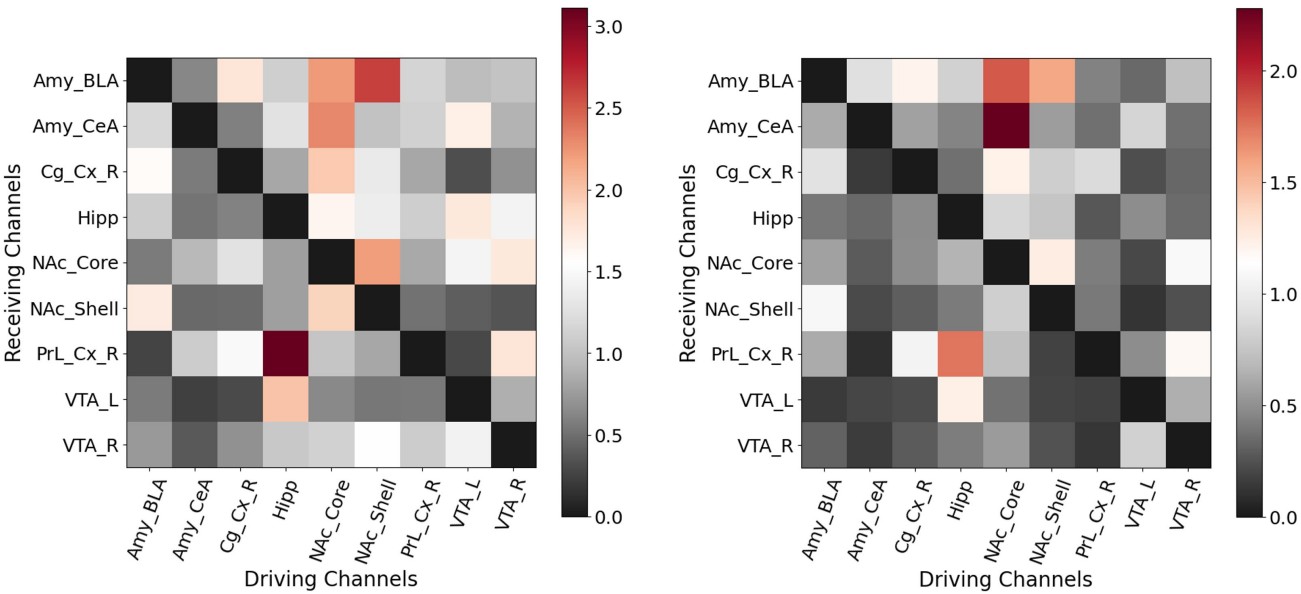

*Figure 32.* Average estimated strength of causal relationships between channels recorded in the Social Preference state from the Region-Averaged SP 100 Hz experiment.

*Figure 33.* Average estimated strength of causal relationships between channels recorded in the Object Preference state from the Region-Averaged SP 100 Hz experiment.

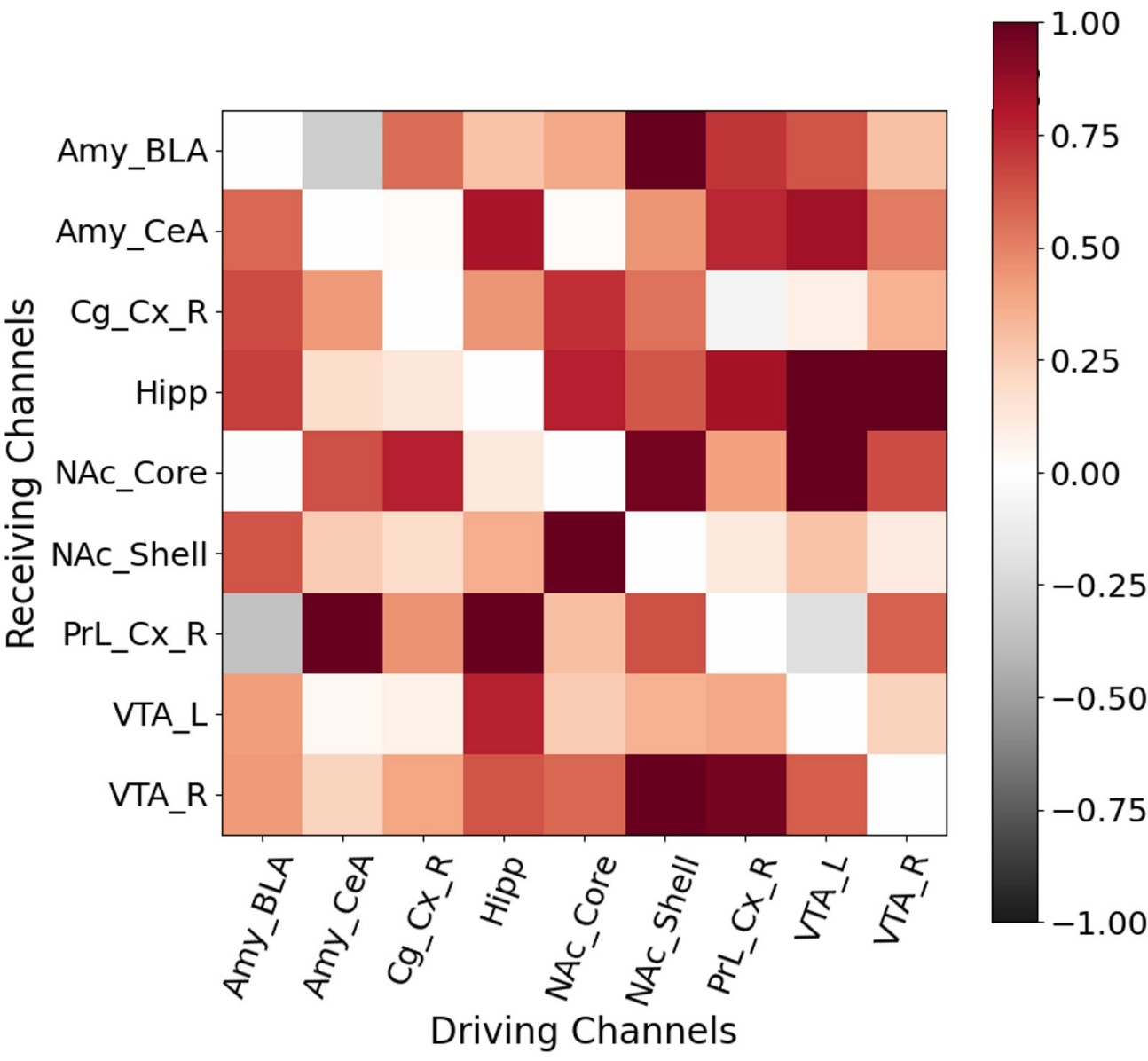

*Figure 34.* Difference in average estimated strength of causal relationships between channels predicted in the Social Preference vs Object Preference behavioral paradigms of the Region-Averaged SP 100 Hz experiment.

# D. Datasets Used: Additional Details

In this section, we report various hyperparameters used in curating/preparing the datasets used in our experiments. Supplemental Table 9 presents hyperparameters for the D4IC dataset (see Section 4.3). Supplemental Table 10 presents hyperparameters for the Synthetic Systems Dataset (see Section 4.2). Supplemental Table 11 presents hyperparameters for the Region-Averaged TST 100 Hz dataset (see Section 5). We briefly discuss the Region-Averaged SP 100 Hz dataset in Appendix D.2.

Note that some elements in these hyperparameter tables appear as functions/equations rather than numerical values or lists of numbers. Due to the vast number of models we had to train for our quantitative results, portions of our code were designed to compute various hyperparameters rather than have them hard-coded and/or passed in as arguments. We release this code in the project repository for the main paper; again, the repository can be accessed at github.com/carlson-lab/redcliff-s-hypothesizing-dynamic-causal-graphs.

*Table 9.* D4IC Dataset (Section 4.3) - hyperparameters across experiments.

| PARAMETER NAME | VALUE |
| --- | --- |
| NETWORK SIZE | 10 |
| NUMBER OF FACTORS PER SYSTEM | 5 |
| NUMBER OF CROSS VALIDATION FOLDS | 5 |
| DOMINANT COEFFICIENT | 10 |
| POSSIBLE BACKGROUND COEFF.S | $\in [0.0, 0.1, 1.0]$ |

*Table 10.* Synthetic Systems Dataset (Sections 4.1-4.2) - hyperparameters across experiments. Of particular note, repeat number 3 (indexed from 0) of the $n_c = 12$-node, $n_e = 33$-edge (inter-variable), $n_k = 3$-factor synthetic system dataset was used to perform grid searches over hyperparameters across our Synthetic Systems experiments. Generally speaking, we find that the greater the number of edges per node in the system, the more 'complex' the Granger causal estimation task.

| PARAMETER NAME | VALUE |
| --- | --- |
| NUMBER OF REPEATS PER SYSTEM | 5 |
| NUMBER OF FACTORS/VAR-MODELS ($n_k = B$) | $n_k \in \{1, 2, 3, 4, 5, 6, 7, 8, 9, 10\}$ |
| DIMENSIONALITY ($n_c$) | $n_c \in \{3, 6, 12\}$ |
| LAGGED DEPENDENCIES ($\tau$) | 2 |
| NUMBER OF TIMESTEPS IN RECORDINGS | 100 |
| TRAINING SAMPLES PER CLASS LABEL (TSCL) | 1040 |
| VALIDATION SAMPLES PER CLASS LABEL (VSCL) | 240 |
| NUMBER OF SAMPLES IN TRAINING SET | $= \text{TSCL} * (B + 1)$ |
| NUMBER OF SAMPLES IN VALIDATION SET | $= \text{VSCL} * (B + 1)$ |
| INNOVATIONS_AMP_COEFFS | $= \text{NP.ONES}((n_c, 1))$ |
| INNOVATIONS_VAR | $= \text{NP.ZEROS}((n_c, 1))$ |
| INNOVATIONS_MU | $= \text{NP.ZEROS}((n_c, 1))$ |
| BASE_FREQUENCIES | $= \text{np.pi} * \text{np.array}([i * 707 + i\%2 \text{ for i in range}(n_c)]).\text{reshape}(n_c, 1)/120000$ |

## D.1. Preprocessing the TST 100 Hz Datasets

As mentioned in the main paper, we applied some preprocessing steps to the publicly available TST dataset (Carlson et al., 2023) to prepare it for our experiments. Specifically, we first sorted the publicly available local field potential (LFP) recordings and associated behavioral label files according to the identifier of the mouse being observed; this was done to ensure all mice were equally represented in each data repeat and so that different mice could be designated as 'holdout' subjects for each repeat. We then iterated through each subject mouse's files and performed the following operations on each recording:

- Removed LFP recordings from any channel/brain-region that was not present across all recordings of all mice;

- Marked any time point for which the LFP value was more than 15 median absolute deviations above the median value of the (filtered - see below) recorded signal as outliers (i.e., 'nan' values);

*Table 11.* Region-Averaged TST 100 Hz Dataset (Section 4.4) - hyperparameters across experiments.

| PARAMETER NAME | VALUE |
|---|---:|
| NUM_PROCESSED_SAMPLES | 10000 |
| SAMPLE_TEMP_WINDOW_SIZE | 1500 |
| SAMPLE_FREQ | 1000 |
| POST_PROCESSING_SAMPLE_FREQ | 100 |
| CUTOFF | 35 |
| MAD_THRESHOLD | 15 |
| Q | 2 |
| ORDER | 3 |
| FILTER_TYPE | LOWPASS |
| HOMECAGE_STOP | = 300*SAMPLE_FREQ |
| DOWNSAMPLING_STEP_SIZE | = SAMPLE_FREQ // POST_PROCESSING_SAMPLE_FREQ |

- Filtered the resulting (outlier-marked) LFPs - where we zero-ed out all outliers - with a low-pass Butterworth filter followed by IIR notch filter;

- Filtered LFP channel recordings were then stacked into an $n_c \times T_{rec}$ recording where $n_c$ was the number of selected LFP channels and $T_{rec}$ was the full length of the recording.

Once a recording file had been run through the above steps, we drew samples evenly across the parts of the recording representing the *a)* Home Cage (HC), *b)* Open Field (OP), and *c)* Tail Suspended (TS) behavioral paradigms. Each sample was drawn in such a way that it was guaranteed to have no marked outlier time points. Once a sample had been drawn, a one-hot vector label corresponding to the behavioral paradigm was attached to it, and the sampled signal was downsampled to reduce computational overhead in later experiments.

### D.2. Preprocessing the SP 100 Hz Datasets

We used the same parameters to preprocess the Social Preference (SP) dataset (Mague et al., 2022) as were used to prepare the Region-Averaged TST 100 Hz dataset. The only fundamental differences between the preprocessed TST data and the Region-Averaged SP 100 Hz dataset were 1) the labels in the datasets (three behavioral labels for TST and two for SP) and 2) the dimensionality of the time series being analyzed.

## E. Model Hyperparameters Used in Experiments: Additional Details

In this section, we report hyperparameters used to configure and train the models used in our experiments. Hyperparameters for the cLSTM models that appear in this paper (see Sections 4.2 and 4.3) are given in Supplemental Table 12. Hyperparameters for the cMLP models that appear in this paper (see Sections 4.2 and 4.3) are given in Supplemental Table 13. Hyperparameters for the DGCNN models that appear in this paper (see Sections 4.2 and 4.3) are given in Supplemental Table 14. Hyperparameters for the DYNOTEARS models that appear in this paper (see Section 4.3) are given in Supplemental Table 15. Hyperparameters for the NAVAR-R (cLSTM) models that appear in this paper (see Section 4.3) are given in Supplemental Table 16. Hyperparameters for the NAVAR-P (cMLP) models that appear in this paper (see Section 4.3) are given in Supplemental Table 17. Hyperparameters for the dCSFA models that appear in this paper (see Sections 4.2 and 4.3) are given in Supplemental Table 18. Hyperparameters for the REDCLIFF-S (cMLP) models that appear in this paper (see Sections 4.2, 4.3, and 5) are given in Supplemental Tables 19 and 20.

### E.1. Identifying Datasets Used to Tune Parameters

Along with the hyperparameters for each model, we report the dataset(s) used to derive those parameters. Models represented by columns with "Synth. Systems" in the column name used hyperparameters selected by grid search(es) on Repeat 3 of the dataset for a (nonlinear, noisy) system with $n_c = 12$, $n_e = 33$, and $n_k = 3$; code for generating this dataset is included in the project repository for the main paper. Models represented by columns with "D4IC" in the column name had their hyperparameters selected by grid search(es) performed on Repeat 0 of the Moderate SNR D4IC dataset (see Table 9), with the cMLP v1 model being less optimal than cMLP v2 according to the original selection criteria.

*Table 12.* cLSTM - hyperparameters across experiments.

| PARAMETER NAME | SYNTH. SYSTEMS | D4IC |
|---|---|---|
| BATCH_SIZE | 128 | 128 |
| MAX_ITER | 300 | 1000 |
| NUM_SIMS | 1 | 1 |
| CONTEXT | 2 | 2 |
| MAX_INPUT_LENGTH | 4 | 4 |
| FORECAST COEFF. | 1.0 | 1.0 |
| ADJ. L1 REGULARIZATION COEFF. | 1.0 | 10.0 |
| GEN_HIDDEN | 25 | 25 |
| GEN_LR | 0.0001 | 0.0005 |
| GEN_EPS | 0.0001 | 0.0001 |
| GEN_WEIGHT_DECAY | 0.0001 | 0.0001 |
| EMBED_HIDDEN_SIZES | 10 | 10 |

*Table 13.* cMLP - hyperparameters across experiments.

| PARAMETER NAME | SYNTH. SYSTEMS | D4IC (V1) | D4IC (V2) |
|---|---|---|---|
| BATCH_SIZE | 128 | 128 | 128 |
| MAX_ITER | 300 | 1000 | 1000 |
| GEN_LAG_AND_INPUT_LEN | 2 | 2 | 2 |
| FORECAST COEFF. | 1.0 | 1.0 | 1.0 |
| ADJ. L1 REGULARIZATION COEFF. | 1.0 | 1.0 | 1.0 |
| GEN_HIDDEN | 25 | 50 | 50 |
| GEN_LR | 0.0001 | 0.0005 | 0.001 |
| GEN_EPS | 0.0001 | 0.0001 | 0.0001 |
| GEN_WEIGHT_DECAY | 0.0001 | 0.0001 | 0.0001 |
| EMBED_HIDDEN_SIZES | 10 | 60 | 10 |

### E.2. Hyperparameters Represented by Functions

We noticed in preliminary experiments that some hyperparameters were highly sensitive to certain aspects of the dynamical systems being modeled; furthermore, we noticed these sensitivities could be described by fairly simple functions of system aspects. Rather than set a fixed value for these model hyperparameters, we tuned 'meta-hyperparameters' (e.g., coefficients) of the functions for these sensitive hyperparameters to a particular dataset, and then recomputed the output of their functions each time the 'tuned' model was transferred to a new target dynamical system. We report the tuned functional form (with tuned meta-hyperparameters) in place of any fixed value for this type of sensitive hyperparameter.

*Table 14.* DGCNN - hyperparameters across experiments.

| PARAMETER NAME | SYNTH. SYSTEMS | D4IC |
|---|---|---|
| BATCH_SIZE | 128 | 128 |
| MAX_ITER | 300 | 1000 |
| NUM_CHANNELS | 10 | 10 |
| GEN_EPS | 0.0001 | 0.0001 |
| GEN_WEIGHT_DECAY | 0.0001 | 0.0001 |
| NUM_FEATURES_PER_NODE | 2 | 2 |
| NUM_CLASSES | = SAME AS DATASET | 5 |
| NUM_GRAPH_CONV_LAYERS | 3 | 1 |
| NUM_HIDDEN_NODES | 250 | 100 |
| GEN_LR | 0.0001 | 0.0001 |

*Table 15.* DYNOTEARS - hyperparameters across experiments.

| PARAMETER NAME | D4IC |
|---|---|
| MAX_ITER | 100 |
| LAMBDA_W | 0.9 |
| LAMBDA_A | 0.1 |
| H_TOL | 0.00000001 |
| W_THRESHOLD | 0.0 |
| TABU_EDGES | NONE |
| TABU_PARENT_NODES | NONE |
| TABU_CHILD_NODES | NONE |
| LAG_SIZE | 1 |

*Table 16.* NAVAR-R (cLSTM) - hyperparameters across experiments.

| PARAMETER NAME | D4IC |
|---|---|
| BATCH_SIZE | 128 |
| EPOCHS | 1000 |
| NUM_NODES | 10 |
| MAXLAGS | 2 |
| NUM_HIDDEN | 256 |
| HIDDEN_LAYERS | 1 |
| LEARNING_RATE | 0.0001 |
| WEIGHT_DECAY | 0 |
| LAMBDA1 | 0.0 |

*Table 17.* NAVAR-P (cMLP) - hyperparameters across experiments.

| PARAMETER NAME | D4IC |
|---|---|
| BATCH_SIZE | 128 |
| EPOCHS | 1000 |
| NUM_NODES | 10 |
| MAXLAGS | 20 |
| NUM_HIDDEN | 256 |
| HIDDEN_LAYERS | 2 |
| LEARNING_RATE | 0.0001 |
| WEIGHT_DECAY | 0 |
| LAMBDA1 | 0.0 |

*Table 18.* dCSFA - hyperparameters across experiments.

| PARAMETER NAME | SYNTH. SYSTEMS | D4IC |
|---|---|---|
| BATCH_SIZE | 128 | 128 |
| N_EPOCHS | 250 | 1000 |
| N_PRE_EPOCHS | 50 | 50 |
| NMF_MAX_ITER | 10 | 20 |
| NUM_HIGH_LEVEL_NODE_FEATURES | 13 | 5 |
| NUM_NODE_FEATURES | 50 | 20 |
| N_COMPONENTS | = SAME AS DATASET | 5 |
| N_SUP_NETWORKS | = N_COMPONENTS | 5 |
| RECON_WEIGHT | 2.0 | 1.0 |
| SUP_WEIGHT | 1.0 | 2.0 |
| SUP_RECON_WEIGHT | 1.0 | 1.0 |
| SUP_SMOOTHNESS_WEIGHT | 1.0 | 2.0 |
| H | 256 | 256 |
| MOMENTUM | 0.9 | 0.5 |
| LR | 0.0005 | 0.001 |
| FS | 100 | 100 |
| MIN_FREQ | 0.0 | 0.0 |
| MAX_FREQ | 50.0 | 50.0 |
| DIRECTED_SPECTRUM | TRUE | TRUE |
| DETREND | CONSTANT | CONSTANT |
| WINDOW | HANN | HANN |
| NPERSEG | = NUM_NODE_FEATURES | = NUM_NODE_FEATURES |
| NOVERLAP | = NUM_NODE_FEATURES//2 | = NUM_NODE_FEATURES//2 |
| NFFT | NONE | NONE |

*Table 19.* REDCLIFF-S - hyperparameters across experiments on synthetic data. Note that for the Synthetic Systems datasets we set $n_c$ equal to the dimensionality of the system, while for the D4IC datasets we had $n_c = 10$.

| PARAMETER NAME | SYNTH. SYSTEMS | D4IC |
|---|---|---|
| BATCH_SIZE | 128 | 128 |
| MAX_ITER | 300 | 1000 |
| $n_k$ | = SAME AS DATASET | 5 |
| $B$ | = $n_k$ | 5 |
| $\tau_{in}$ (EQ. 2) | 4 | 4 |
| $\omega$ (EQ. 5) | 10.0 | 10.0 |
| $\rho$ (EQ. 5) | $= \frac{1.0}{\sum_{i}^{n_k-1} i}$ | $0.1 = \frac{1.0}{\sum_{i}^{n_k-1} i}$ |
| $\eta$ (EQ. 5) | $= \frac{\frac{0.1}{0.1}}{n_k\sqrt{n_c{}^2-1}}$ | $0.00201 \approx \frac{\frac{0.1}{0.1}}{n_k\sqrt{n_c{}^2-1}}$ |
| $\gamma$ (EQ. 6) | 0.001 | 0.001 |
| $\lambda$ (EQ. 9) | 100.0 | 100.0 |
| GEN_HIDDEN | [25] | [100] |
| GEN_LR | 0.0005 | 0.0005 |
| GEN_EPS | 0.0001 | 0.0001 |
| GEN_WEIGHT_DECAY | 0.0001 | 0.0001 |
| NUM_PRETRAIN_EPOCHS | 100 | 50 |
| NUM_ACCLIMATION_EPOCHS | 100 | 15 |
| FACTOR_SCORE_EMBEDDER_TYPE | DGCNN | DGCNN |
| EMBED_NUM_HIDDEN_NODES | 100 | 30 |
| EMBED_NUM_GRAPH_CONV_LAYERS | 3 | 2 |
| EMBED_LR | 0.0005 | 0.0002 |
| EMBED_EPS | 0.0001 | 0.0001 |
| EMBED_WEIGHT_DECAY | 0.0001 | 0.0001 |
| EMBED_LAG ($\tau_{in} + \tau_{class}$) | 16 | 20 |

*Table 20.* REDCLIFF-S - hyperparameters across experiments on real-world data. Note that for the Region-Averaged TST 100 Hz dataset we had $n_c = 12$ while for the Region Averaged SP dataset we had $n_c = 9$.

| PARAMETER NAME | REG. AVG. TST 100 HZ | REG. AVG. SP 100 HZ |
|---|---|---|
| BATCH_SIZE | 128 | 128 |
| MAX_ITER | 300 | 300 |
| $n_k$ | 9 | [2, 4, 6, 9, **18**, 36] |
| $B$ | 3 | 2 |
| $\tau_{in}$ (EQ. 2) | 4 | 4 |
| $\omega$ (EQ. 5) | 10.0 | 10.0 |
| $\rho$ (EQ. 5) | $= \frac{1.0}{\sum_{i=1}^{n_k-1} i} \approx 0.0278$ | $= \frac{1.0}{\sum_{i=1}^{n_k-1} i}$ |
| $\eta$ (EQ. 5) | $= \frac{0.1}{n_k \sqrt{n_c^2-1}} \approx 0.000929$ | $= \frac{0.1}{n_k \sqrt{n_c^2-1}}$ |
| $\gamma$ (EQ. 6) | 0.001 | 0.001 |
| $\lambda$ (EQ. 9) | 100.0 | 100.0 |
| GEN_HIDDEN | [25] | [25] |
| GEN_LR | 0.0005 | 0.0005 |
| GEN_EPS | 0.0001 | 0.0001 |
| GEN_WEIGHT_DECAY | 0.0001 | 0.0001 |
| NUM_PRETRAIN_EPOCHS | 100 | 100 |
| NUM_ACCLIMATION_EPOCHS | 100 | 100 |
| FACTOR_SCORE_EMBEDDER_TYPE | DGCNN | DGCNN |
| EMBED_NUM_HIDDEN_NODES | 100 | 100 |
| EMBED_NUM_GRAPH_CONV_LAYERS | 3 | 3 |
| EMBED_LR | 0.0005 | 0.0005 |
| EMBED_EPS | 0.0001 | 0.0001 |
| EMBED_WEIGHT_DECAY | 0.0001 | 0.0001 |
| EMBED_LAG ($\tau_{in} + \tau_{class}$) | 16 | 16 |

