# OpenReview forum: "Generating Hypotheses of Dynamic Causal Graphs in Neuroscience: Leveraging Generative Factor Models of Observed Time Series"
_ICML.cc/2025/Conference — ICML 2025 poster_

### Official Review · Reviewer_GHbs · 2025-02-25

**Overall Recommendation:** 4

**Summary:**

The authors propose a factor model-based method to generate possible Granger causal hypotheses about time series data given by a time-dependent (i.e., non-static) data-generating process. The generated hypotheses are adjacency matrices and summarize possible causal effects among the observed system variables and their history. Intuitively, the proposed method learns a set of generative factor functions that predict the behavior of the time series/system given a history of $\tau$ values as well as a state model that predicts a weighting vector of factors given said history. The factors are parameterized as neural networks where the first layer is interpreted as an adjacency matrix showing Granger causal relationships among variables while the state model is a neural network predicting the relevance of each factor given a history (thus introducing dynamic behavior and allowing to "mix" adjacency based on the model state).
Empirically, the proposed method seems to improve upon existing baselines on synthetic and real-world tasks.

**Claims And Evidence:**

The paper claims that the proposed method generates hypotheses (i.e., Granger causal graphs) for nonlinear and state-dependent time series systems, thereby improving upon existing baselines that have stricter assumptions such as linearity or static-ness of the system.

By design, the method should be able to accurately capture nonlinear dependencies among system variables due to the use of neural networks. Also, an explicit state model is learned, representing the state of the modeled time series system and weighting a set of learned factor functions which include a representation of the Granger causal adjacency matrix. Combining these two components, the proposed method's design allows the modeling of nonlinear dependencies **and** model state-dependent time series systems.

The claims are supported by empirical evidence in the experimental section. The authors first test their method on artificially generated data that exhibits nonlinear dependencies among variables and state-dependent behavior.

**Essential References Not Discussed:**

The approach is closely related to the method proposed in [1]. In [1], the authors propose to learn dynamic causal graphs from data using neural ordinary differential equations, thereby modeling temporal dependencies among variables.

**References**

[1] Cheng et al. DyCAST: Learning Dynamic Causal Structure from Time Series. ICLR 2025.

> Note: The work in [1] was published in 2025. Thus, there is no need to incorporate it as an additional baseline, in my opinion. Nevertheless, it should be cited.

**Experimental Designs Or Analyses:**

The experimental design is valid and sound. However, I am not familiar with neuroscience literature. Therefore, it is hard to tell whether the DREAM4 and TST datasets are appropriate choices to test the capabilities of the proposed method.

**Methods And Evaluation Criteria:**

The authors explicitly aim to construct a model that can be used for hypothesis generation in neuroscience applications. Hence, their evaluation of a real-world brain data problem is appropriate.

**Other Comments Or Suggestions:**

-

**Other Strengths And Weaknesses:**

**Strengths**
- the paper is generally clearly written
- leveraging factor-based time series models to extract adjacency matrices is novel and innovative
- the modeling choices are well-motivated
- the authors test their method on real-world brain data and show significant improvements

**Weaknesses**
- Fig. 1 is too large and should be placed on the top of the page to improve readability
- a rigor definition of "optimal f1 score" is missing (see questions) (Sec. 4.2)
- it is unclear what the authors mean by "pairwise improvement" in Fig. 2

**Questions For Authors:**

- with how many seeds were the experiments conducted?
- it is unclear what the authors mean by "pairwise improvement" in Fig. 2B. Which metric is considered here exactly? Why is lower better than higher?
- what is the "optimal f1-score"? Could the authors provide an exact definition?

**Relation To Broader Scientific Literature:**

As discussed by the authors, the work is mainly related to [1], [2], and [3], covering causal discovery and its application to neuroscience. In [1], the authors aim to discover causal models with nonlinear dependencies using local linear approximations. This restricts the expressiveness of the learned model. [2] and [3] discuss the necessity of using state-dependent models in modeling brain data and demonstrate the usefulness of modeling brain data causally.

**References**

[1] Fujiwara et al. Causal Discovery for Non-stationary Non-linear Time-series Data Using Just-In-Time Modeling. ICLR 2023.

[2] Mague et al. Brain-wide electrical dynamics encode individual appetitive social behavior. Neuron 2022.

[3] Talbot et al. Estimating a brain network predictive of stress and genotype with supervised autoencoders. Journal of the Royal Statistical Society. 2023.

**Theoretical Claims:**

The authors claim and show that:
- nonlinear, dynamic causal graphs are unidentifiable in general (App. A1)
- assuming non-trivial noise, a window of size $\tau$ is not sufficient to distinguish two (or more) factors if the classifier is assumed to be unbiased (App. A2)

Although the authors show in A2 that for a time-window $T < \tau$ it is impossible to have an accurate and unbiased estimator to distinguish between $f_i$ and $f_j$, it is not shown that having $T > \tau$ is sufficient to achieve that goal.

---

> ### Author Rebuttal · Authors · 2025-04-01
>
> We are very grateful for the reviewer's thoughtful feedback and suggestions on how we can further improve our paper. We respond to their concerns below.
>
> Question 1: "with how many seeds were the experiments conducted?"
>
> Each dataset was curated using a fixed random seed ('9999' for Synthetic Systems, '1337' for TST, '5' for D4IC) to ensure data could be generated/replicated on other devices (see provided code). With regards to model training, we used a single random seed ('0') throughout our experiments to ensure reproducibility, which can be seen in the included code (esp. at the beginning of "_ _ main _ _" in each of our python-based training files). In concurrence with best practices, these seeds were chosen arbitrarily so as to avoid 'seed'-hacking.
>
> Question 2: "it is unclear what the authors mean by 'pairwise improvement' in Fig. 2B. Which metric is considered here exactly? Why is lower better than higher?"
>
> We thank the reviewer for giving us the opportunity to clarify our terminology. 'Pair-wise' here refers to the practice of matching predictions made by pairs of algorithms on the same graph/system and evaluating the differences. In the case of Fig. 2B, 'pair-wise improvement' is the difference between the optimal f1-score obtained by REDCLIFF-S (say, $f1_{redc}$) and the optimal f1-score obtained by another algorithm (say, $f1_{base}$) on a given graph $g$; that is, $\mathrm{PWImprovement} = f1_{redc}(g) - f1_{base}(g)$. Thus, the higher the value, the better REDCLIFF-S did relative to the given baseline. In theory, the pair-wise improvement could take on any value between -1 (if REDCLIFF-S had an f1-score of 0 and the baseline's was 1) and 1 (with REDCLIFF-S' f1-score being 1 and the baseline being at 0).
>
> Question 3: "what is the 'optimal f1-score'? Could the authors provide an exact definition?"
>
> The reviewer rightly points out an area which needs clearer discussion in our final draft. In essence, the f1-score compares positive predictions with positive labels, but this requires that a 'positive prediction' threshold be set for each algorithm. Rather than pick a threshold which may skew prediction results in our favor, we instead used a different threshold for each algorithm on each prediction such that the algorithm/baseline attained the highest possible f1-score performance on that prediction (i.e. the 'optimal' f1-score). Our current draft discusses the 'optimal f1-score' at the end of the second paragraph in Section 4 of our paper, in which we state that "we compute each ... algorithm's f1-score using a classification threshold ... which yields the highest possible f1-score for that algorithm", but we will absolutely expound on this in order to improve our paper. Are there any particular key points that the reviewer would like us to include in our final draft beyond what we have described above?
>
> Concern 1: Location of Figure 1
>
> We thank the reviewer for their suggestion to improve the readability of our paper, and will adjust Fig. 1 so that it is located at the top of the page.
>
> Concern 2: Additional References
>
> We thank the reviewer for calling our attention to the DyCAST paper, and will certainly include it in our discussion of related works.
>
> Again, we thank the reviewer for their feedback and hope our response has been helpful.

---

### Official Review · Reviewer_xCcy · 2025-03-14

**Overall Recommendation:** 3

**Summary:**

This work introduces a deep generative factor model that uses weighted superposition of static graphs to achieve dynamic causal graph modeling and capture nonlinear relationships. For complex neural activity in the brain, the method does well in detecting time-varying interactions between neural variables. Through evaluation on synthetic datasets, this work shows that the proposed model achieves higher performance compared to leading methods.

**Claims And Evidence:**

A large number of experimental results on synthetic datasets demonstrate the effectiveness and uniqueness of the approach in this work. However, there are few experiments on real neural datasets (only one), and a lack of comparisons with other baseline models. Evaluation and comparison on more real datasets (e.g. the dataset used in [1]) could be more convincing of the model's contribution to the field of neuroscience.

[1] Mague, S. D., et al. Brain-wide electrical dynamics encode individual appetitive social behavior. Neuron. 2022.

**Essential References Not Discussed:**

Please see the above comment of **Relation To Broader Scientific Literature**.

**Experimental Designs Or Analyses:**

As discussed in the comment of **Claims And Evidence**, more experiments on real neural datasets may help to strengthen the claim and contribution of this work.

**Methods And Evaluation Criteria:**

The use of dynamic coefficients to control the efficacy of multiple static models is novel to the field, although this practice is already common in machine learning. The use of behavioral information to guide dynamic coefficients is constructive and helpful. The objective function of the model is well-designed and theoretically supported.

This work uses the f1-score as the primary metric. Although the authors have claimed that ROC-AUC is less appropriate in this situation, I think this work should also report ROC-AUC results, as this metric is widely used in related work.

**Other Comments Or Suggestions:**

N/A

**Other Strengths And Weaknesses:**

1. There is a lack of ablation studies for the method, such as an unsupervised version of REDCLIFF ($\lambda$=0 in equation 9), a model without dynamic factors (coefficients are the same for all factors), a single-factor model, and a model without the cosine similarity penalty.
2. The settings of the sections need to be fine-tuned. Section 5 is a discussion of Sections 4.1 to 4.3, which should be included as a subsection in Section 4. Section 6 is the experiment on a real-world dataset and is juxtaposed with Section 4. Therefore, it would be preferable to change the title of Section 4 to "Experiments on simulated datasets" and Section 6 to "Experiments on real-world datasets".

**Questions For Authors:**

1. As there are many hyperparameters in the method, how do the authors perform the parameter search? Does it cause excessive training costs? How do these hyperparameters affect the model's performance?

**Relation To Broader Scientific Literature:**

The key contribution of this work is to capture causal relationships between neural variables in the brain. The authors discuss broad scientific literature, including linear and nonlinear causal discovery methods, dynamic graph modeling and some applications of causal discovery in neuroscience. Some works [2-3] have proposed dynamical systems to extract low-dimensional latent dynamics from neural activity, although they did not discover causal relationships. It would be better to include these related works for discussion.

[2] Linderman, S., et al. Bayesian Learning and Inference in Recurrent Switching Linear Dynamical Systems. In Proceedings of the 20th International Conference on Artificial Intelligence and Statistics. 2017.

[3] Keshtkaran, M. R., et al. A large-scale neural network training framework for generalized estimation
of single-trial population dynamics. Nature Methods. 2022.

**Theoretical Claims:**

This work describes an accurate mathematical modeling of the method. A detailed discussion and derivation of the model design is also presented, which provides helpful theoretical support.

---

> ### Author Rebuttal · Authors · 2025-04-01
>
> Question 1 (Parameter selection and training cost):
>
> Most of our hyperparameters were chosen via grid-search, though we did identify some simple equations relating the $\rho$ and $\eta$ parameters in Eq. 5 to other parameters in the model and/or dataset that we used to adapt those specific parameters to new settings (see Appendix E.2 and corresponding Table 13). We discuss how the number of parameters affects our analyses in the first paragraph of Appendix B.4 - essentially, the high number of parameters in the REDCLIFF-S model does increase training costs and did restrict us to tuning hyperparameters on a subset of the Synthetic Systems datasets. However, we emphasize that there were 51 different systems each with 5 different repeats that we sought to use in those experiments (48 of which made it into the paper and/or appendices, see Appendix Figure 9).
>
> Question 2 (Ablation Studies):
>
> Following the reviewer's suggestion, we performed ablations on several REDCLIFF-S parameters across multiple systems. We will include the full extent of these experiments in the final paper, and report the following results obtained on our D4IC HSNR system (Sec. 4.3) here - given as the average (Avg) Optimal F1 score across repeats and factors and corresponding standard error of the mean (SEM):
>
>  - Ablation: $\rho=0$ (Eq. 5 & 10) :: 0.304 (Avg) 0.006 (SEM)
>  - Ablation: $n_k=1$ (Eq. 2 equivalent) :: 0.316 (Avg) 0.011 (SEM)
>  - Ablation: $\alpha=1$ (Eq. 4) :: 0.338 (Avg) 0.009 (SEM)
>  - Ablation: $\lambda=0$ (Eq. 9) :: 0.332 (Avg) 0.006 (SEM)
>
> Note each ablation resulted in significantly worse performance than that reported in Figure 4 of our paper, verifying the utility of each parameter.
>
> Concern 1: Paper Organization
>
> We thank the reviewer for raising valid feedback that will improve the readability of our paper; we will move Section 5 into Section 4 as a subsection and rename the section headings as suggested in the final paper.
>
> Concern 2: Related Works
>
> We thank the reviewer for suggesting several related works for us to include in our paper, and we will be sure to incorporate them in our final draft.
>
> Concern 3: Additional Real-world (Neural) Datasets
>
> As suggested by the reviewer, we ran an additional case study on the Social Preference dataset first used in [1]. We describe the results briefly here and will release a more detailed discussion of our findings in our final draft. We used a random subset of 20 out of the 28 mice for model cross validation, leaving the remaining 8 mice as holdouts. Regarding parameter selection, we borrowed the majority of parameters from our TST case study (see Appendix Table 13), with a few changes to reflect the Social Preference dataset. Specifically, we adjusted the number of supervised factors from 3 to 2 and assigned one to "Social Preference" label and the second factor to the "Object Preference" label. We then performed a 5-fold cross-validated grid search on the number of total factors, which resulted in the selection of the 18-factor model(s).
>
> As with the TST case study, we averaged the predicted factors to arrive at our final predictions. We find the difference between the Social Preference and Object Preference factors resembles:
>
>  - VTA_L -> Hipp
>  - VTA_L -> NAc_Core
>  - NAc_Core <--> NAc_Shell
>  - NAc_Shell -> Amy_BLA
>  - NAc_Shell -> VTA_R
>  - VTA_R -> Hipp
>  - Hipp -> PrL_Cx_R
>  - PrL_Cx_R -> VTA_R
>
> Interestingly, the Hippocampus (Hipp) features prominently in REDCLIFF-S' predicted factor difference, whereas it does not seem as relevant in [1]. Given the complexities of studying sociability in mice and the significant differences between REDCLIFF-S and the methods used in [1], the discrepancies in these findings may be highlighting entirely different aspects of social behavior.
>
>  - [1] Mague, S. D., et al. Brain-wide electrical dynamics encode individual appetitive social behavior. Neuron. 2022.
>
> Concern 4: Inclusion of ROC-AUC
>
> The reviewer points out ROC-AUC is more popular in related works, and we will certainly include ROC-AUC performance as supplementary information in our final draft. Due to limited space, we simply note our analyses of our paper's '6-2-2', '6-4-2', and '12-11-2' Synthetic Systems (Figure 2A) suggests the reported Opt. F1 score performances largely reflect the ROC-AUC scores obtained using the same thresholded predictions (with the exception of DCSFA-NMF in the 6-4-2 system, which seems to perform similarly to REDCLIFF-S in terms of ROC-AUC, despite having a lower mean Opt. F1 score). As discussed regarding Reviewer Qgsb's Question 2, our Synthetic Systems labels omit information regarding the presence of 'background' factor activity, which limits the examples of 'true negative'/inactive relationships between nodes in our dataset and may reduce the applicability of ROC-AUC scores generally, whereas the F1-score more accurately reflects our data generation process.
>
> We thank the reviewer again for their feedback, and hope they found our response helpful.

---

### Official Review · Reviewer_Qgsb · 2025-03-16

**Overall Recommendation:** 3

**Summary:**

This paper proposes a novel hypothesis generation framework for dynamic causal graphs in neuroscience, aiming to improve the efficiency of causal discovery in time-dependent systems. The authors introduce REDCLIFF-S, a model designed to estimate state-dependent causal interactions by combining:
	1.	Factor-based models for dynamic causal graphs.
	2.	Supervised auxiliary variables (e.g., behavioral labels) to refine hypotheses.
	3.	A flexible generative framework that adapts causal interactions over time.

**Claims And Evidence:**

The claims made in the submission are supported by convincing evidence in general.

Potential Weaknesses in Claims:
* No analysis on how the model generalizes beyond neuroscience.
* No formal guarantees on identifiability

**Essential References Not Discussed:**

No comparison to deep learning-based causal discovery approaches (e.g., DAG-GNN, Transformer-based models) and kernel-based method (e.g., CDNOD).

**Experimental Designs Or Analyses:**

Strengths
* Ablation study on synthetic systems shows clear performance trends (Figure 2).
* DREAM4 dataset adaptation demonstrates the model’s effectiveness across different complexity levels.

Weaknesses
* No ablation study on the impact of supervised auxiliary labels.

**Methods And Evaluation Criteria:**

Yes. The used datasets and evaluation metrics make sense for the problem. The used datasets include synthetic datasets (vector autoregressive models, nonlinear causal graphs), DREAM4 Insilico-Combined (D4IC) dataset (biological system benchmarks), and TST (Tail Suspension Test) neural recordings (real-world case study). Evaluation metrics include F1 score of causal graph estimation (Figures 2, 4), Computational efficiency (Table 1), and Qualitative validation with neuroscientific findings (Figure 5).


Limitations
* No evaluation on datasets outside neuroscience (e.g., economics, social systems).
* No analysis of model sensitivity to hyperparameters or noise.

**Other Comments Or Suggestions:**

No

**Other Strengths And Weaknesses:**

Strengths
* Novel hypothesis generation for dynamic causal graphs.
* Improves causal discovery accuracy in neuroscience applications.
* Empirical validation on synthetic and real-world data.

Weaknesses
* No discussion on generalizability beyond neuroscience.
* Limited theoretical discussion on identifiability.

**Questions For Authors:**

The paper introduces dynamic causal graphs via factor models, but does not discuss formal identifiability conditions. Are there any theoretical guarantees on when REDCLIFF-S produces unique causal structures? Does the factorization step introduce ambiguities in the inferred causal graphs?

The supervised extension (REDCLIFF-S) incorporates behavioral labels for refining causal inference. How much do behavioral labels contribute to improved performance? Would REDCLIFF-S still outperform unsupervised methods if auxiliary labels were unreliable or noisy?

**Relation To Broader Scientific Literature:**

This paper extends causal discovery in time series (Runge, 2020; Gerhardus & Runge, 2020), generalizes dynamic graph models beyond PCMCI+ and DYNOTEARS, and ntroduces state-dependent causal inference relevant to neuroscience.

**Theoretical Claims:**

I did not check the proofs.

There is no formal guarantee on the identifiability.

---

> ### Author Rebuttal · Authors · 2025-04-01
>
> We are very grateful for the reviewer's thoughtful feedback. We respond to their concerns below.
>
> Question 1: "Are there any theoretical guarantees on when REDCLIFF-S produces unique causal structures? Does the factorization step introduce ambiguities in the inferred causal graphs?"
>
> As we allude to in our introduction and expound upon in Appendix A.1, it is very difficult to prove identifiability for even the simplest nonlinear, state dependent systems (our example consists of 2 nodes A and B where A drives B according to 2 different functions depending on the global state). Thus, we focus our efforts on rigorous empirical testing across numerous dynamical systems (48 unique configurations each with 5 different repeats in our Synthetic Systems experiments - see Figure 9 in Appendix C.3).
>
> As the reviewer rightly points out, our focus on emprical evidence may mean we miss ambiguities in our proposed REDCLIFF-S algorithm. However, due to the structure of our algorithm, these ambiguities are mostly related to the relative importance of each unsupervised factor as opposed to the relationship between a given factor and the input and output data (put another way, each factor has clear linear relationships to the input and output data - see Equation 4 as well as Appendix B.1 - but the weighting of each factor does not). For this reason, we outline our selection criteria in Equation 10 and demonstrate how we use it to restrict the number of unsupervised factors to the minimal amount necessary in our TST Case study in Section 6 (see also Figure 25 of the Appendices).
>
> Question 2: "How much do behavioral labels contribute to improved performance? Would REDCLIFF-S still outperform unsupervised methods if auxiliary labels were unreliable or noisy?"
>
> This is an important question, and one which we should address more directly in our paper. In short, all of the Synthetic Systems experiments we present in this paper contain label noise, as do the D4IC MSNR and LSNR experiments presented in Appendix C.2. With regard to curating the labels for our Synthetic Systems experiments, we mention in the second paragraph of Section 4.1 that "Recordings obtained from each VAR model were then weighted over time by linearly interpolated weights (randomly selected between 0 and 1). These weighted recordings were added together along with a level of Gaussian noise. We augmented labels by marking with a one-hot vector which factor weight was largest at each time step". Thus, all of our labels in the Synthetic Systems experiments occluded information relating to how active each system was at a given time step. While we do not quantify how much information was occluded from the Synthetic Systems labels, we do provide a simpler exploration of this between our D4IC HSNR, MSNR, and LSNR datasets. As discussed in Section 4.3, the D4IC experiments each had "down-weighted recordings from all but one" system factor added on top of the recording from a "main" system factor (the identity of which determined the label for a given sample). This background noise was was weighted with a scalar coefficient valued at either 0.0, 0.1, or 1.0 while the recording of the 'main' system factor was multiplied by 10. As can be seen in Appendix C.2 (please forgive the typo in the caption of Figure 8, it should read 'D4IC-MSNR' instead of 'D4IC-HSNR') our proposed algorithm does not perform well under this particular form of noise. We believe that this is due to the fact that there was no variation/interpolation between the weight applied to each 'background' factor, making it difficult to ultimately distinguish them from the 'main' factor.
>
> We thank the reviewer for mentioning additional literature that we can discuss in our paper, and we will certainly include it in our final draft. We hope our response has been helpful, and thank the reviewer again for their feedback.

---

### Official Review · Reviewer_nDzu · 2025-03-18

**Overall Recommendation:** 3

**Summary:**

The submission propose a method for causal discovery (at least this is what I understand).
Summarizing it seems that the proposed approach is another implementation of a non-linear Granger causality model with some particular choice of implementing architecture and loss function.
There are no theoretical results (identifiability, consistency etc..), instead the authors argue that such systems are non-identifiable in the appendix, and why it is still valuable.
The proposed approach is benchmarked against other approaches with a simulation experiment.

**Claims And Evidence:**

partially, in general there are no strong theoretical claims.

Nevertheless, I have concerns about:

(1) "The main strength of REDCLIFF-S is that its hypotheses can be easily tested" (first sentence in discussion) what "tested" mean ? is it possible to do statistical inference ? why other methods hypothesis can't be tested? to my understanding the problem of post-selection inference in causal discovery is far from solved, except using the naive data-splitting, I am only aware of a Greedy Equivalent Search (GES) variant which allow for valid inference https://arxiv.org/pdf/2208.05949  .

(2) In the conclusions:  "attains state-of-the-art performance in estimating multiple causal graphs simultaneously in various settings, and in a way conducive to follow-up scientific inquiry"  where is the evidence of that? what have been measured to prove that in the simulation or the real world data? Moreover  I think multiple baselines are missing, see next.

**Essential References Not Discussed:**

I think they main literature not discussed is the Dynamic Causal Model approach which was developed exactly for causal discovery in Neuroscience.

see for instance
- Friston, Karl J., Lee Harrison, and Will Penny. "Dynamic causal modelling." Neuroimage 19.4 (2003): 1273-1302.
- Friston, Karl J., et al. "Dynamic causal modelling of COVID-19." Wellcome open research 5 (2020): 89.
- Stephan, Klaas Enno, et al. "Ten simple rules for dynamic causal modeling." Neuroimage 49.4 (2010): 3099-3109.
- Friston, Karl, Rosalyn Moran, and Anil K. Seth. "Analysing connectivity with Granger causality and dynamic causal modelling." Current opinion in neurobiology 23.2 (2013): 172-178.

for a comprehensive recent reviews you can refer to:

- Discovering causal relations and equations from data
Gustau Camps-Valls, Andreas Gerhardus, Urmi Ninad, Gherardo Varando, Georg Martius, Emili Balaguer-Ballester, Ricardo Vinuesa, Emiliano Diaz, Laure Zanna, and Jakob Runge  Physics Reports 1044   https://www.sciencedirect.com/science/article/pii/S0370157323003411

- Causal Discovery from Temporal Data: An Overview and New Perspectives   https://dl.acm.org/doi/10.1145/3705297#sec-3

- Review of Causal Discovery Methods Based on Graphical Models https://www.frontiersin.org/journals/genetics/articles/10.3389/fgene.2019.00524/full

**Experimental Designs Or Analyses:**

correct

**Methods And Evaluation Criteria:**

Yes the proposed data and benchmark makes sense.
I would suggest the addition of some causality aware metrics between the learned graphs, apart from the F1 score.

for instance:

- "Structural intervention distance for evaluating causal graphs"  https://dl.acm.org/doi/10.1162/NECO_a_00708
- "Adjustment Identification Distance: A gadjid for Causal Structure Learning"  https://openreview.net/forum?id=jO5UNNrjJr
- " Separation-Based Distance Measures for Causal Graphs"  https://openreview.net/forum?id=KO7fATqh2W

**Other Comments Or Suggestions:**

see above


——-

Score updated after rebuttal

**Other Strengths And Weaknesses:**

The problem is not really clearly stated, for instance in sec. 3.1. Problem Statement I would expect a clear statement on which is the inference goal.

**Questions For Authors:**

- What is clearly the goal, recovering a causal graph?
- what are the assumptions ?, of course different sets of assumptions would lead to different algorithms and methods
- How the proposed approach compare to Dynamic Causal Models?
- Why PCMCI style algorithms where not compared ? what about simpler baselines such as  tidybench https://github.com/sweichwald or a simple structural VAR?
-

**Relation To Broader Scientific Literature:**

I think numerous baselines are missing
for instance PCMCI variants are not considered in the experiments and in the Related Work they are partially dismissed because
"However, these methods tend to rely on assumptions that rule out hypotheses we would like to explore in neuroscience; for example, PCMCI+, LPCMCI, and DBNs all assume causal graphs are acyclical" they assume the complete graph (expanded in time) is acyclic, they do not assume that there are no feedback loops in time for instance.

Moreover some literature is not discussed (see next question).

**Theoretical Claims:**

none

---

> ### Author Rebuttal · Authors · 2025-04-01
>
> Question 1: Our Goal
>
> Our goal is to reduce the space of hypotheses that must be tested before a causal relationship is successfully identified. We allude to this in our abstract, but will make this more explicit in our final draft.
>
> Question 2: Our Assumptions
>
> We only assume our data consists of regularly sampled time series of scalar-valued nodes (see Section 3.1). These time series may be noisy, and we make no assumptions regarding the underlying generative processes / graphs. In Sec. 3.5, we assume the availability of "global state" labels for each time step that relate to the dominant generative process.
>
> Question 3: Comparing REDCLIFF-S & Dynamic Causal Models
>
> The REDCLIFF-S method should be viewed as a preparatory analysis technique preceding the implementation of a more sophisticated method such as Dynamic Causal Models (DCMs). As discussed in work cited by the reviewer [1,2], the Granger causal paradigm assumed by REDCLIFF-S precludes the availability of interventional inputs required to construct a DCM model [1], suggesting that Granger causal models (such as REDCLIFF-S) be employed as a complementary analysis prior to the implementation of DCMs [2].
>
>  - [1] pages 3101 & 3102 of Stephan, Klaas Enno, et al. "Ten simple rules for dynamic causal modeling." Neuroimage 49.4 (2010): 3099-3109.
>
>  - [2] page 175 of Friston, Karl, Rosalyn Moran, and Anil K. Seth. "Analysing connectivity with Granger causality and dynamic causal modelling." Current opinion in neurobiology 23.2 (2013): 172-178.
>
> Question 4: PCMCI-style algorithms & simpler baselines (e.g. tidybench https://github.com/sweichwald)
>
> We originally excluded PCMCI and similar methods over concerns with acyclicity constraints, particularly related to cross-frequency coupling phenomena we hoped to capture. Given the reviewer's feedback, we compared RECLIFF-S, Regime-PCMCI [3], and supervised versions of 'slarac', 'qrbs', and 'lasar' from the tidybench repository ('selvar' did not compile locally) on our '12-11-2' Synthetic Systems (Figure 2A), with results shown below. Values are given as the average (Avg) of the mean statistic of system factors across repeats and the standard error of the mean (SEM) for continuous-valued metrics or as the median (Med) statistic across repeats and factors for discrete metrics; 'Upper' and 'Lower' refer to the fact that we had to employ causal distance metrics [4] on the upper-triangular and lower-triangular portions of our adjacency matrices due to the presence of cycles (details in Sec. 4.1 & Appendix B.2).
>
> REDCLIFF-S
>  - Opt. F1: 0.373 (Avg) 0.028 (SEM)
>  - ROC-AUC: 0.687 (Avg) 0.017 (SEM)
>  - Upper Ancestor Aid Errors: 3.0 (Med)
>  - Lower Ancestor Aid Errors: 2.0 (Med)
>  - Upper Parent Aid Errors: 4.5 (Med)
>  - Lower Parent Aid Errors: 2.5 (Med)
>  - Upper Oset Aid Errors: 3.0 (Med)
>  - Lower Oset Aid Errors: 3.0 (Med)
>
> Regime-PCMCI
>  - Opt. F1: 0.407 (Avg) 0.027 (SEM)
>  - ROC-AUC: 0.673 (Avg) 0.026 (SEM)
>  - Upper Ancestor Aid Errors: 4.0 (Med)
>  - Lower Ancestor Aid Errors: 4.0 (Med)
>  - Upper Parent Aid Errors: 6.0 (Med)
>  - Lower Parent Aid Errors: 5.5 (Med)
>  - Upper Oset Aid Errors: 4.0 (Med)
>  - Lower Oset Aid Errors: 4.0 (Med)
>
> (Supervised) slarac
>  - Opt. F1: 0.417 (Avg) 0.034 (SEM)
>  - ROC-AUC: 0.598 (Avg) 0.015 (SEM)
>  - Upper Ancestor Aid Errors: 6.0 (Med)
>  - Lower Ancestor Aid Errors: 4.5 (Med)
>  - Upper Parent Aid Errors: 8.0 (Med)
>  - Lower Parent Aid Errors: 6.5 (Med)
>  - Upper Oset Aid Errors: 6.0 (Med)
>  - Lower Oset Aid Errors: 4.5 (Med)
>
> (Supervised) qrbs
>  - Opt. F1: 0.451 (Avg) 0.029 (SEM)
>  - ROC-AUC: 0.614 (Avg) 0.014 (SEM)
>  - Upper Ancestor Aid Errors: 6.1 (Med)
>  - Lower Ancestor Aid Errors: 5.0 (Med)
>  - Upper Parent Aid Errors: 8.9 (Med)
>  - Lower Parent Aid Errors: 7.0 (Med)
>  - Upper Oset Aid Errors: 6.1 (Med)
>  - Lower Oset Aid Errors: 5.0 (Med)
>
> (Supervised) lasar
>  - Opt. F1: 0.283 (Avg) 0.009 (SEM)
>  - ROC-AUC: 0.563 (Avg) 0.026 (SEM)
>  - Upper Ancestor Aid Errors: 5.3 (Med)
>  - Lower Ancestor Aid Errors: 4.7 (Med)
>  - Upper Parent Aid Errors: 8.5 (Med)
>  - Lower Parent Aid Errors: 6.7 (Med)
>  - Upper Oset Aid Errors: 5.7 (Med)
>  - Lower Oset Aid Errors: 4.7 (Med)
>
> Note Regime-PCMCI and REDCLIFF-S attain significantly higher ROC-AUC scores than the other baselines. While the Opt. F1 scores for REDCLIFF-S are similar to several baselines', we highlight REDCLIFF-S' lower median error across all of the causal metrics shown as evidence that it offers improvements in identifying functional relations in this setting.
>
>  - [3] Elena Saggioro, Jana de Wiljes, Marlene Kretschmer, Jakob Runge; Reconstructing regime-dependent causal relationships from observational time series. Chaos 1 November 2020; 30 (11): 113115. https://doi.org/10.1063/5.0020538
>
>  - [4] "Adjustment Identification Distance: A gadjid for Causal Structure Learning" https://openreview.net/forum?id=jO5UNNrjJr
>
> We thank the reviewer for their helpful feedback which we will certainly incorporate into our paper, and we hope they found our response helpful.

---

> > ### Comment · Reviewer_nDzu · 2025-04-01
> >
> > I thank the author for the rebuttal.
> > I will raise my score accordingly.

---

### Decision · Program_Chairs · 2025-05-01

**Decision:**

Accept (poster)

**Comment:**

This paper presents a novel deep generative factor model for dynamic causal graph discovery, designed to address the challenges of modeling non-linear and state-dependent relationships in time series data, particularly within the context of neuroscience. The reviewers generally agree that the paper presents a promising approach with clear potential, supported by empirical results on both synthetic and real-world (though limited) datasets.

Reviewers raised concerns regarding missing experimental details, literature, and comparison and a lack of assessment of variability (e.g., with random seeds).

However, the strengths – an innovative method, a well-motivated approach, and demonstrated promising results – warrant acceptance contingent on the authors addressing the reviewers’ comments, particularly by providing better experimental details, including additional baseline experiments (e.g. in response to reviewer nDzu) and additional variability experiments.